EMBO
Molecular Medicine

# 7D, a small molecule inhibits dengue infection by increasing interferons and neutralizing-antibodies via CXCL4:CXCR3:p38:IRF3 and Sirt1:STAT3 axes respectively

Kishan Kumar Gaur [iD] [1,4], Tejeswara Rao Asuru [iD] [1,4], Mitul Srivastava[2], Nitu Singh[1], Nikil Purushotham[3], Boja Poojary[3], Bhabatosh Das[2], Sankar Bhattacharyya [iD] [2,5], Shailendra Asthana [iD] [2,5 ✉] & Prasenjit Guchhait [iD] [1,5 ✉]

## Abstract

**There are a limited number of effective vaccines against dengue virus (DENV) and significant efforts are being made to develop potent anti-virals. Previously, we described that platelet-chemokine CXCL4 negatively regulates interferon (IFN)-α/β synthesis and promotes DENV2 replication. An antagonist to CXCR3 (CXCL4 receptor) reversed it and inhibited viral replication. In a concurrent search, we identified CXCR3-antagonist from our compound library, namely 7D, which inhibited all serotypes of DENV in vitro. With a half-life of ~2.85 h in plasma and no significant toxicity, 7D supplementation (8 mg/kg-body-weight) to DENV2-infected IFNα/ β/γR$^{-/-}$AG129 or wild-type C57BL6 mice increased synthesis of IFN-α/β and IFN-λ, and rescued disease symptoms like thrombocytopenia, leukopenia and vascular-leakage, with improved survival. 7D, having the property to inhibit Sirt-1 deacetylase, promoted acetylation and phosphorylation of STAT3, which in-turn increased plasmablast proliferation, germinal-center maturation and synthesis of neutralizing-antibodies against DENV2 in mice. A STAT3-inhibitor successfully inhibited these effects of 7D. Together, these observations identify compound 7D as a stimulator of IFN-α/β/λ synthesis via CXCL4:CXCR3:p38:IRF3 signaling, and a booster for neutralizing-antibody generation by promoting STAT3-acetylation in plasmablasts, capable of protecting dengue infection.**

**Keywords** CXCR3-antagonist; CXCL4; Dengue; Interferons; Antibodies
**Subject Categories** Immunology; Microbiology, Virology & Host Pathogen Interaction

## Introduction

Viral infections are a major public health concern on a global scale. The ability of viruses to mutate rapidly remains the major hurdle in developing effective pharmaceutics against these simple nucleic acid entities enveloped by protein. The recent COVID-19 pandemic has given impetus to the efforts being made towards developing anti-virals that can restrict viral transmission at the initial stage. Indeed, recent advisory of the World Health Organization encourages research and development of anti-virals against several diseases, including dengue. Dengue, caused by DENV a positive-sense single-stranded RNA virus of family Flaviviridae, is now endemic to more than 100 countries including India. There has been a considerable rise in the incidence of the disease worldwide in recent years, increasing from 505,430 cases in 2000 to 5.2 million in 2019 (Bhatt et al, 2013). A modelling report estimates about 390 million infections annually; ~96 millions of these infections have clinical implications in 128 countries (Brady et al, 2012).

Dengue symptoms usually appear 4–10 days post-infection and last for 2–7 days. Apart from asymptomatic infection the clinical manifestations of the disease include pyrexia of unknown origin and serious complications like dengue haemorrhagic fever (DHF) and dengue shock syndrome (DSS). Severe dengue is associated with thrombocytopenia, plasma leakage and complications of coagulopathy (Kalayanarooj, 2011; Simmons et al, 2012). All four serotypes (DENV1, DENV2, DENV3 and DENV4) share sequence homology but possess distinct immunoreactivity; thus, when secondary infections with different serotypes occur after primary infection with one serotype, the likelihood of severe dengue infection increases. This is due to a process known as antibody-dependent enhancement (ADE) of infection, in which the neutralizing antibodies from first infection can bind to the next invading DENV of another serotype, facilitating their entry into the monocytes via Ig-Fc receptor interaction. (Littaua et al, 1990;

[1]Regional Centre for Biotechnology, National Capital Region Biotech Science Cluster, Faridabad, Haryana, India. [2]Translational Health Science Technology Institute, National Capital Region Biotech Science Cluster, Faridabad, Haryana, India. [3]Department of Studies in Chemistry, Mangalore University, Mangalagangotri, Karnataka, India. [4]These authors contributed equally: Kishan Kumar Gaur, Tejeswara Rao Asuru. [5]These authors contributed equally as senior authors: Sankar Bhattacharyya, Shailendra Asthana, Prasenjit Guchhait. ✉E-mail: sasthana@thsti.res.in; prasenjit@rcb.res.in

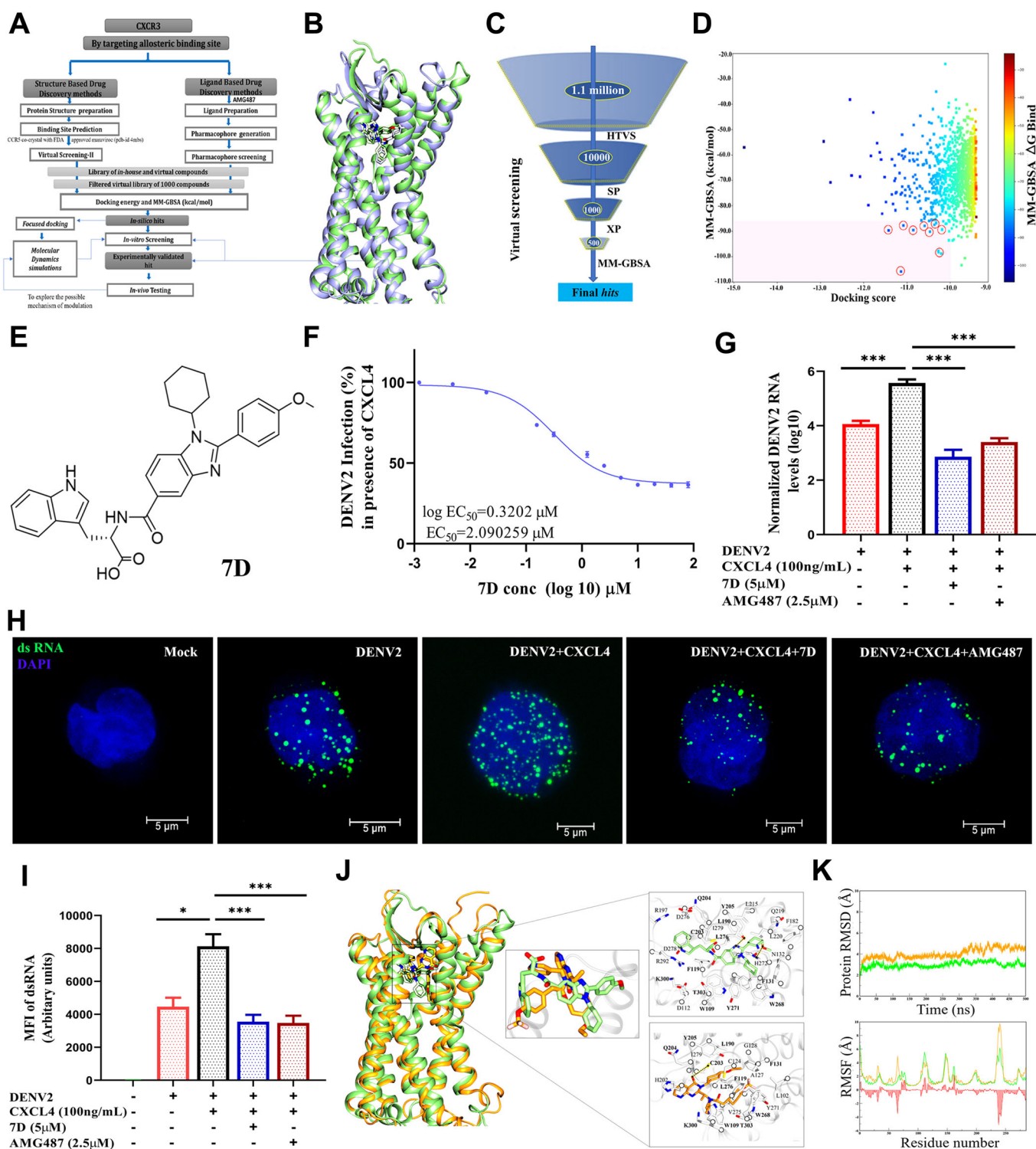

Dejnirattisai et al, 2010; Guzman et al, 2013). Additionally, progression of secondary dengue infection is also exacerbated by virus-antibody complexes leading to complement activation and T cell-mediated immune responses (Prince et al, 2011; Cucunawangsih et al, 2015). Serotypic antibody cross reactivity and subsequent enhancement of infection have impeded the development of effective vaccines against DENV.

Several cytokines and chemokines play important roles in immunoregulation and pathogenesis of dengue. CCL2 (MCP1) increases DENV infection whereas CCL5 (RANTES) restricts it (Sierra et al, 2014). Low CCL5 and high CXCL8 (IL8) in plasma in initial stages of dengue infection have been studied as predictive markers of severe dengue (Patra et al, 2019). CXCL4, primarily released by activated platelets (Gleissner et al, 2008), is reported to

**Figure 1. Screening CXCR3 antagonists and testing their anti-viral effects in vitro.**

(A–E) Screening CXCR3 antagonists. (A) Computational pipeline to identify CXCR3 antagonists. (B) Superimposed structures docked state of CXCR3 (green)-compound "7D" and crystal CCR5 (blue)- maraviroc (MRV). 7D and MRV are shown in licorice representation and colored in atom-wise C: purple/green, N: blue, O: red and F: Pink. (C) In silico virtual screening. High throughput virtual screening (HTVS)>Standard-precision (SP) docking>Extra-precision (XP) docking> Molecular mechanics with generalized born and surface area solvation (MM-GBSA). (D) Binding free energies of molecules vs docking energies. 13 hit-molecules (dot represents a molecule) were circled in red. (E) Molecular structure of 7D. (F–I) Testing anti-viral effect of 7D. (F) U937-DC-SIGN cells were infected with MOI ~ 1 of DENV2 (strain P23085 INDI-60) in presence CXCL4 (100 ng/ml) for 24 h, and viral genome was quantified using qRT-PCR. The $EC_{50}$ value of 7D was calculated from the dose-response curve from independent experiments, $n = 3$. (G) DENV2 genome were measured using qRT-PCR from the above experiment, $n = 3$ independent experiments, one-way ANOVA and Bonferroni's post-test were used ($P$ values: 0.001; 0.001; 0.001). (H) Viral dsRNA (green) was measured using microscopy. (I) Data are the mean fluorescence intensity (MFI), $n = 3$ independent experiments, (40 cells per group), Kruskal–Wallis test was used ($P$ values: 0.03; 0.0008; 0.0006). (J, K) In silico binding of 7D or AMG487 with CXCR3. (J) Superimposed docking states of CXCR3-7D (green) and docked state of CXCR3-AMG487 (yellow). The inset shows the binding mode of 7D and AMG487. (K) Dynamical characteristics of systems were elucidated through MD simulations. Root-mean square deviation (RMSD) evolution through the course of 500 ns. Ligand RMSD of docked pose of CXCR3-7D (green) and docked pose of CXCR3-AMG487 (yellow). The Cα atomic fluctuation observed in all systems CXCR3-7D and CXCR3-AMG487 are represented by RMSF. Data information: (F, G, I) Data are mean ± SEM, *$P < 0.05$, ***$P < 0.001$. Source data are available online for this figure.

be abundant in the plasma of dengue patients. Studies suggest that it promotes replication of the DENV virus in immune cells, including monocytes (Trugilho et al, 2017; Fragnoud et al, 2015). Our earlier study has described that CXCL4 binding to receptor CXCR3 increases p38 phosphorylation and decreases interferon (IFN)-regulatory factor (IRF)-9 expression, in turn inhibits IFNα synthesis (Ojha et al, 2019). Besides, others have described the protective roles of other chemokines like CXCL9 and CXCL10 in dengue mediated via a common receptor CXCR3 (Hsieh et al, 2006; Ip and Liao, 2010). Although we did not explore the role of CXCL9/ CXCL10, we did describe that the binding of CXCL4 to CXCR3 suppresses IFNα synthesis in dengue infection in vitro, and supplementation with a CXCR3-antagonist, namely AMG487, reverses it (Ojha et al, 2019). Unfortunately, the therapeutic usage of AMG487 have been limited following its withdrawal from the Phase II clinical trial against rheumatoid arthritis due to limited efficacy.

To develop an alternative anti-viral, we screened CXCR3 antagonists and identified compound "7D" from our in-house library as a promising candidate for inhibiting DENV replication. With a half-life of 2.85 h in plasma and almost no cytotoxic effects, the administration of 7D to DENV2-infected mice rescued symptoms like thrombocytopenia and vascular leakage, and improved animal survival. Like AMG487, 7D improved synthesis of IFNs, but unlike the former, it increased DENV2-neutralizing antibodies and antigen-specific B lymphocyte percentage in DENV-infected mice. These observations suggest 7D as a potent therapeutic against dengue infection.

## Results

### Compound 7D bound CXCR3 and inhibited CXCL4-mediated DENV replication in monocytes

In order to develop CXCR3 antagonists, we constructed 3D structure of CXCR3 using homology modelling of CCR5 (Fig. EV1A–C), since CXCR3 crystal structure is unavailable. We screened ~1.1 million compounds virtually, and selected approximately 13 molecules showing considerable binding (MM-GBSA score) to CXCR3 (Fig. 1A–D). One of these compounds "7D [2-(1-cyclohexyl-2-(4-methoxyphenyl)-1H-benzo[d]imidazole-5-carbox-amido)-3-(1H-indol-3-yl) propanoic acid, Mol. Wt. 563]" (Fig. 1E) from our in-house library (Purushotham et al, 2022), was found to

be potentially inhibiting DENV2 replication in monocytic U937-DC-SIGN cells in vitro with an $EC_{50}$ ~2.1 μM (Fig. 1F). Other 12 compounds did not show inhibitory effects against DENV2 (Fig. EV1D–F). 7D (5 μM) suppressed the CXCL4-mediated enhancement of DENV2 replication (Fig. 1G–I), similar to the effect of AMG487 ($C_{32}H_{28}F_3N_5O_4$) or benzeneacetamide, N-[(1 R)-1-[3-(4-ethoxyphenyl)-3,4-dihydro-4-oxopyrido[2,3-d]pyrimidin-2-yl]ethyl]-N-(3-pyridinylmethyl)-4-(trifluoromethoxy), Mol. Wt. 634, ($EC_{50}$ ~2.41 μM, Fig. EV1D). We described the anti-DENV2 effect of AMG487 in our earlier work as well (Ojha et al, 2019). In fact, the molecular dynamics simulations of 7D showed thermo-dynamically stable binding to CXCR3 as compared to AMG487 (Fig. 1J,K), which is supported by binding energy of interacting amino acid residues with either 7D or AMG487 (Fig. EV1C). 7D did not show cytotoxic effect in MTT assay in vitro (Fig. EV1E). Importantly, this small molecule inhibited the replication of all other serotypes DENV1, DENV3 and DENV4 in vitro (Fig. EV1G–I).

### 7D inhibited CXCL4-mediated DENV2 replication via CXCR3 axis in vitro

We confirmed the role of CXCL4:CXCR3 axis in 7D-mediated inhibition of viral infection using monocytes isolated from CXCR3$^{-/-}$ or CXCR3$^{+/+}$ (WT) mice. 7D suppressed the CXCL4-mediated DENV2 (strain EU081177.1, Rathore et al, 2021) replication in CXCR3$^{+/+}$ monocytes. Conversely, the DENV2 replication in CXCR3$^{-/-}$ monocytes was found unaltered in presence of either CXCL4 or 7D in vitro (Fig. 2A–I), suggesting the involvement of CXCL4:CXCR3 axis in 7D-mediated inhibition of DENV replication. Even the expression of downstream-adapter molecule of CXCL4:CXCR3 signaling pathway like p38MAPK, a regulator of IFNα/β synthesis, was found unaltered in CXCR3$^{-/-}$ monocytes in vitro from the above experiment (Fig. 2F,G,J,K). Thus, confirming the role of CXCR3 axis in 7D-mediated inhibition of DENV2 replication in monocytes in vitro.

Besides, we also tested the effects of 7D in CXCL4-expressing cells including mouse primary megakaryocytes and human megakaryoblast cell line MEG-01. 7D inhibited the DENV2 infection in both cell types (Fig. EV2B,F). These cells secreted CXCL4 in supernatant following DENV2 infection (Fig. EV2A,E). Further, 7D was unable to inhibit viral infection in cells like primary monocytes of mice or human DC-SIGN-U937 cell line (Fig. EV2D,H) that do not secrete elevated CXCL4 (Fig. EV2C,G).

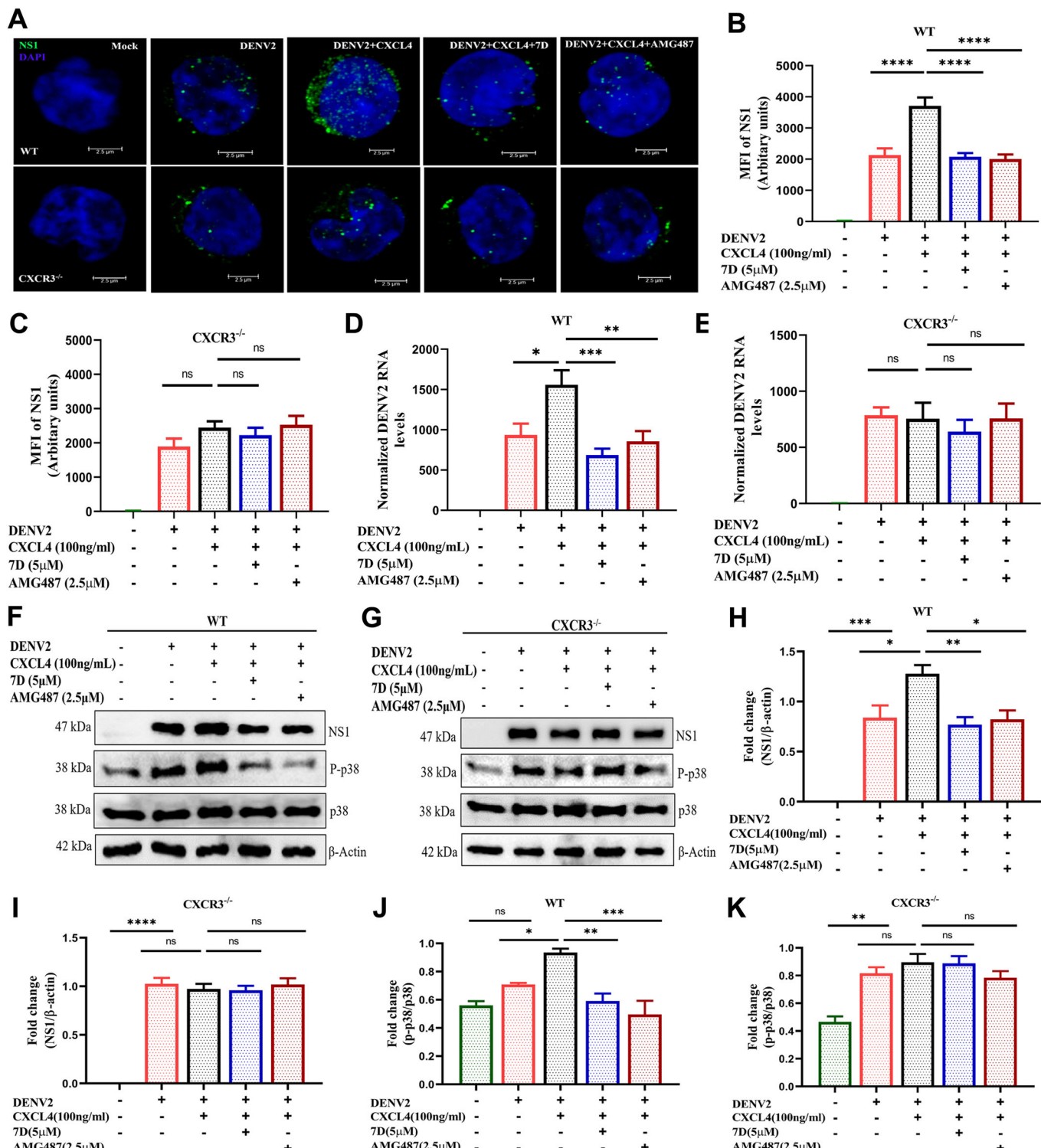

This observation supports the inhibitory effect of 7D on CXCL4-induced DENV2 replication.

Initially, 7D was synthesized as an inhibitor to Sirtuin-1 (Sirt-1) that deacetylates target molecules like p53 (Purushotham et al, 2022). We therefore tested the role of 7D:Sirt1 axis in DENV replication in vitro in the above experimental condition. 7D treatment increased the expression of acetylated (Ac)-p53 in CXCR3$^{-/-}$ monocytes without altering the expression of DENV2 NS1 protein (Fig. EV1J–L). Further, we performed the above experiment in Sirt1$^{-/-}$ mouse embryonic fibroblasts (MEFs) and observed the inhibitory effects of 7D on DENV2 replication in both Sirt1$^{-/-}$ and WT MEFs (Fig. EV3B–M). The 7D supplementation increased the protein acetylation (Ac) in WT but not in Sirt1$^{-/-}$ MEFs, thus, ruling out a direct involvement of Sirt1 axis in DENV2 replication in vitro.

**Figure 2. 7D inhibits viral replication in monocytes by inhibiting CXCR3 receptor.**

(A–E) 7D inhibits viral replication in CXCR3$^{+/+}$ monocytes but not in CXCR3$^{-/-}$. (A) Monocytes collected from peripheral blood of WT C57BL/6 (CXCR3$^{+/+}$) or CXCR3$^{-/-}$ mice were infected with MOI ~1 DENV2 (EU081177.1 strain) in presence of CXCL4 (100 ng/ml) and AMG487 (2.5 μM) or 7D (5 μM). (B, C) After 24 h, cells were processed for NS1 staining using microscopy and data are presented as MFI, $n = 3$ independent experiments, (P values: 0.0001; 0.0001; 0.0001). (D, E) DENV2 genome was quantified from above experiment using qRT-PCR. $n = 3$ independent experiments, (P values: 0.01; 0.0006; 0.005). (F–K) (F–G) Western blot analysis was performed for dengue NS1, P-p38, normalized to β-actin. (H–K) Densitometry data of the above blots, $n = 3$ independent experiments, (P values: H: 0.001; 0.01; 0.005; 0.01, I: 0.0001, J: 0.03; 0.003; 0.0005, K: 0.002). Data information: (B–E, H–K), one-way ANOVA and Bonferroni's post-test were used. Data are mean ± SEM, *$P < 0.05$, **$P < 0.01$, ***$P < 0.001$. ****$P < 0.0001$, ns non-significant. Source data are available online for this figure.

## 7D increased IFN synthesis via CXCR3 axis in vitro

Further, we tested the effect of 7D on CXCL4:CXCR3:p38:IRF:IFN axis. Like AMG487, the compound 7D also improved the expression of IFN-α/β (*IFNA1* and *IFNB1*) and IFN-stimulated genes including *TRIM69* in U937-DC-SIGN cells (Fig. 3A–C) in conjunction with decreased phosphorylation (P) of p38 and increased P-IRF3 (Fig. 3D–G). As mentioned above, 7D is a Sirt-1 inhibitor. We therefore tested the effects of 7D on CXCL4:CXCR3:p38:IRF:IFN axis in Sirt1$^{-/-}$ MEFs and observed an improved secretion of IFNα alongside increased P-IRF3 (Fig. EV3B–M).

## 7D improved dengue symptoms and mice survivability

We developed dengue infection model in AG129 (IFNα/β/γR$^{-/-}$) mice. DENV2 (strain P23085 INDI-60) was adapted in AG129 mice and tested for infection and dengue disease symptoms like thrombocytopenia, leukopenia, vascular leakage and decreased body weight (Appendix Fig. S1A–F). The substitution of nucleotides in envelop and non-structural (NS) protein regions, increased the infectivity of this mouse adapted DENV2 strain, mentioned details in Appendix Table S1. Before testing the anti-DENV effects of 7D, we examined that this compound has no significant cytotoxic effect on liver, kidney and blood cells in mice. HPLC analysis of mouse plasma showed a half-life of $T_{1/2}$ ~2.85 h with a $C_{max}$ ~126 μg/ml after single dose of 7D (8 mg/kg body weight) administered to mice (Appendix Fig. S2A–K). Supplementation with 7D for 4 days improved mice survivability from 15% at 5 DPI to 60% at 15 DPI (Fig. 4B), and body weight in DENV2-infected mice (Fig. 4C). The DENV2-infected mice showed elevation in CXCL4 level in plasma, which was unaltered after 7D treatment (Fig. 4D). The 7D significantly decreased DENV2 replication in liver and spleen of these mice (Fig. 4E,F), and improved platelet and monocyte, but not neutrophil counts in peripheral blood, which were decreased after DENV2 infection (Fig. 4G–I). Besides, the thrombo-inflammatory markers including platelet-leukocyte aggregates and cytokines IL6, TNFα and IL1β in peripheral blood were decreased in DENV2-infected mice after 7D treatment (Fig. 4J–M). The clinical manifestation of severe dengue like vascular leakage was significantly decreased in organs including liver and spleen in infected mice after 7D treatment (Fig. 4N,O). 7D supplementation also increased IFN synthesis (Fig. 4P–R) and antibody generation (Fig. 5) in these mice.

## 7D increased IFNs in infected mice

The 7D treatment increased the levels of IFNα and IFNβ, but not IFNγ in DENV2-infected AG129 mice plasma (Fig. 4P–R). The AG129 mice lack IFNα/β/γR, we therefore confirmed the above observation of 7D-mediated enhancement of IFNα and IFNβ

expression in WT C57BL6 mice infected with another DENV2 (strain EU081177.1). 7D treatment to DENV2-infected WT mice showed the similar trend of increase in *Ifna1* (IFNα) and *Ifnb1* (IFNβ) gene expression (Fig. 6), and IFNα and IFNβ levels in plasma (Fig. 6; Appendix Fig. 3G–I), suggesting a potential therapeutic role of this compound. As expected, infected WT C57BL6 mice showed a milder pathogenicity of dengue infection as compared to AG129 mice. This is because of the anti-viral response of IFNα/β in WT mice, which was blunted in IFNα/β/γR$^{-/-}$AG129 mice. Independent of infection models, 7D increased the IFNα/β concentration in mice plasma.

Further, we examined the expression of type III IFNλ in bone marrow derived macrophages (BMDMs) from IFNα/β/γR$^{-/-}$AG129 (Fig. EV4) as well as WT (Appendix Fig. S6) mice. Another study has described the anti-viral effect of type-III IFNs under the influence of IRF3 transcription in virus infected cells (Wack et al, 2015). Our study described that a blocking antibody to IFNλ2/3 suppressed the expression of interferon-stimulated genes (ISGs) and increased DENV2 infection in these BMDMs (Fig. EV4A–E), indicating a possible anti-viral role of type-III IFN in IFNα/β/γR$^{-/-}$ AG129 mice. 7D supplementation increased the IFNλ levels via P-IRF3 signaling in DENV2-infected BMDMs (Fig. EV4F–L).

## 7D increased STAT3 acetylation and proliferation of plasmablasts, and increased germinal center formation and antibody synthesis, and improved DENV2 neutralization in infected mice

The 7D treatment to DENV2-infected AG129 mice increased the levels of DENV2-neutralizing antibodies both IgM (at 3 DPI) and IgG (at 6 DPI) in circulation (Fig. 5A–D). Importantly, 7D treatment increased the percentage of plasma cells in spleen of infected mice (Fig. 5E). ELISPOT analysis showed the increased number of DENV2-specific plasma cells in spleen of the infected mice after 7D treatment (Fig. 5F,G). Furthermore, we describe that elevated acetylation (Ac-STAT3$^{K685}$) as well as phosphorylation of STAT3 (P-STAT3$^{Y705}$) in B-lymphocytes in spleen of 7D-treated mice (Fig. 5H–M), which could be because of the inhibitory role this compound on Sirt1 deacetylase (Purushotham et al, 2022). That could be a reason for elevated proliferation of the plasmablasts. These 7D-treated mice exhibited increased GL7 expressing follicles indicating a developed germinal center formation (Fig. 5N,O). However, 7D treatment to DENV2-infected WT C57BL6 mice didn't alter the antibody levels in circulation (Fig. 6H,I). We confirmed the involvement of STAT3 signaling in plasmablast proliferation and antibody generation. Treatment with Stattic, a STAT3-inhibitor, suppressed the phosphorylation but not acetylation of STAT3 in lymphocytes of 7D-treated infected mice,

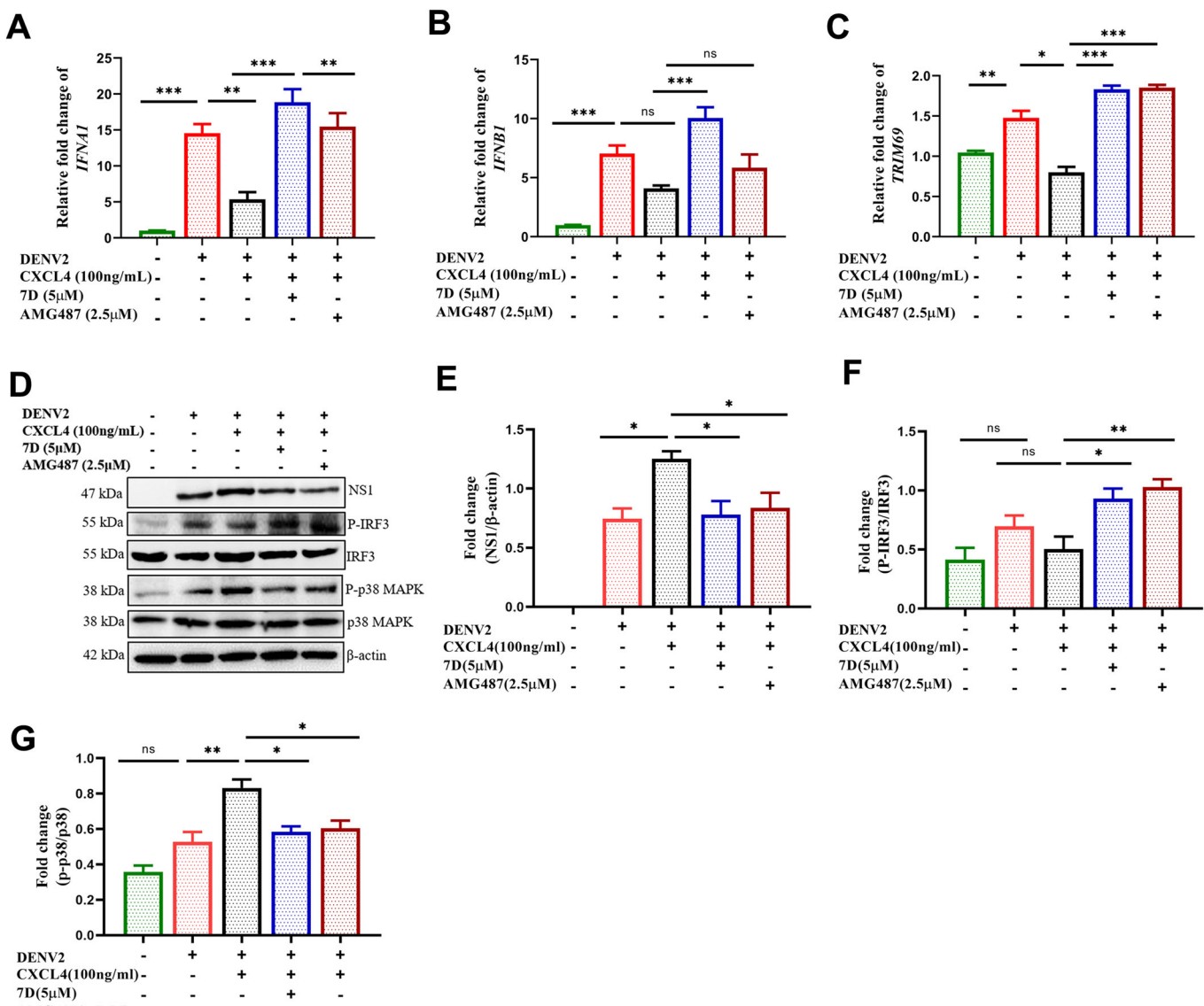

**Figure 3. Effect of 7D on interferon synthesis in vitro.**

(A–C) IFNA1, IFNB1 and TRIM69 genes were measured using qRT-PCR from cell pellets from the above experiments Fig. 1G–I, $n = 3$ independent experiments, (P values: **A**: 0.0002; 0.003; 0.0002; 0.0016, **B**: 0.0006; 0.0007, **C**: 0.002; 0.03; 0.0002; 0.0004). (D–G) Cell pellets were processed for western blot analysis of DENV NS1, P-IRF3 and P-p38, normalized to β-actin. (E–G) Densitometry of the above blots, $n = 3$ independent experiments, (P values: **E**: 0.01; 0.02; 0.02, **F**: 0.03; 0.009, **G**: 0.002; 0.01, 0.01). Data information: (A–C, E–G), one-way ANOVA and Bonferroni's post-test were used. Data are mean ± SEM, *$P < 0.05$, **$P < 0.01$, ***$P < 0.001$, ns non-significant. Source data are available online for this figure.

and decreased the IgG levels in plasma (Fig. EV5). It also suggests that acetylation (Ac-STAT3$^{K685}$) probably promotes phosphorylation of STAT3 (P-STAT3$^{Y705}$). However, the above observation does not explain the decrease in both viral load as well as neutralizing antibodies against DENV2 in Stattic-treated mice. It may be attributed to a global anti-viral effect of this STAT3-inhibitor which needs further investigations.

We also tested the effect of AMG487 in DENV2-infected AG129 mice. The AMG487 (8 mg/kg body weight) supplementation for 4 days decreased the viral replication in spleen and liver at 6 DPI. But unlike 7D, it was unable to increase the DENV2-specific antibody levels in mice (Fig. EV6A–I).

Furthermore, we examined the effects of 7D on DENV2 infection in CXCR3$^{-/-}$ mice. 7D supplementation inhibited the viral replication in spleen of these mice (Fig. 6A). Although, it did not rescue the IFNα levels (Fig. 6C), it improved DENV2-netralizing antibody levels in the plasma (Fig. 6C,F,G). An opposing observation was noted in DENV2-infected WT mice after 7D treatment; 7D increased IFNα (Fig. 6D) but not the antibody levels (Fig. 6E,H,I) in plasma of WT mice. 7D increased the acetylation (Ac-STAT3$^{K685}$) as well as phosphorylation of STAT3 (P-STAT3$^{Y705}$) in B-lymphocytes (Appendix Fig. S3A–C), suggesting a stimulatory effect of this small molecule on plasmablast proliferation and antibody generation in CXCR3$^{-/-}$ mice.

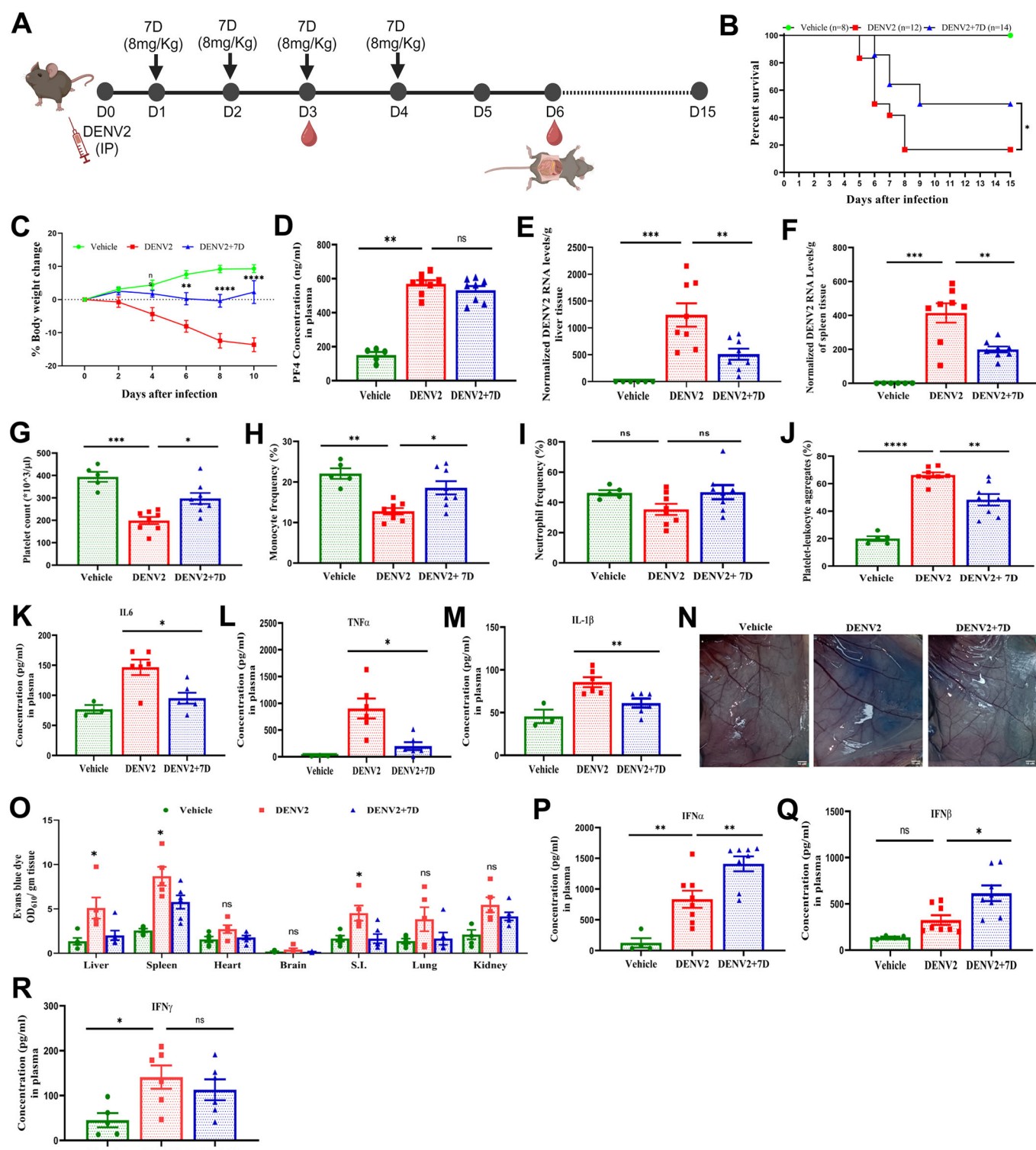

## 7D inhibited DENV2 infection in human monocytes in vitro

We further validated the anti-viral effect of 7D in human monocytes in vitro. 7D (5 μM) significantly suppressed the PF4-mediated enhancement of DENV2 replication in PBMCs

(microscopy of viral dsRNA, Appendix Fig. S6A,B; and qRT-PCR assays of viral genome copies, Appendix Fig. S7C). Likewise, the 7D treatment rescued the expression of *IFNA1/IFNB1* (IFNα/β) genes but not (*IFNG*) IFNγ in infected human monocytes (Appendix Fig. S7D–F), suggesting an inhibitory effect of this compound on DENV infection in human leukocytes.

Figure 4. 7D improves dengue disease pathology and mice survivability alongside increases IFN synthesis in AG129 mice.

(A) Schematic representation of mice experiment. AG129 mice were intraperitonially (i.p.) infected with $10^5$ FFU mouse-adapted DENV2 virus (P8- P23085 INDI-60) or incomplete L15 media as mock. 7D (8 mg/kg body weight) was administered (i.p.) till 4 days post-infection (DPI) and other half were injected with vehicle (PBS + 10% Tween 80) and euthanized at 6 DPI for following assays. (B) In a similar experiment, mice were observed till 15 DPI to obtain survival curves using the Kaplan–Meier method. The curve comparison analysis was performed between the DENV2 ($n = 12$) and DENV2 + 7D ($n = 14$). Log-rank (Mantle Cox) test was used for statistical analysis (P value: 0.03). (C) Change in the body weight was recorded till 15 DPI, $n = 7$ vehicle, $n = 8$ mice per group, two-way ANOVA was used for data analysis, (P values: 0.009; 0.0001; 0.0001). (D) Plasma CXCL4 levels from above mice was measured using ELISA at 6 DPI, $n = 5$ vehicle, $n = 8$ mice per group, one-way ANOVA and Kruskal–Wallis test were used, (P value: 0.005). (E, F) DENV2 genome was quantified by qRT-PCR in (E) liver and (F) spleen tissues, respectively, $n = 6$ vehicle, $n = 8$ mice per group, one-way ANOVA and Bonferroni's post-test were used, (P values: E: 0.001; 0.004, F: 0.001; 0.005). (G–M) (G) Platelet, (H) monocytes, (I) neutrophil and (J) platelet-leukocyte aggregates were measured from peripheral blood of mice from above experiment using flow cytometry, $n = 5$ vehicle, $n = 8$ mice per group, one-way ANOVA and (G, H: Kruskal–Wallis test) and (I, J: Bonferroni's post-test) were used, (P values: G: 0.0005; 0.03, H: 0.003; 0.03, J: 0.0001; 0.005). Gating strategy is mentioned in Appendix Fig. S4A. Plasma levels of (K) IL6, (L) TNFα and (M) IL1β were measured using CBA assay, $n = 3$ vehicle, $n = 6$ mice per group, Mann–Whitney U test was used, (P values: K: 0.02; L: 0.03; M: 0.004). (N, O) Vascular leakage in mice blood vessels. (N) Image of Evan's blue dye extravasation from the veins in abdominal region of DENV2-infected mice. (O) Quantification of Evan's blue dye in different tissues, $n = 5$ vehicle, $n = 5$ for DENV2 and, $n = 6$ for DENV2 + 7D, two-way ANOVA was used, (P values: 0.02; 0.03; 0.04). (P–R) Plasma (P) IFNα, (Q) IFNβ, $n = 4$ vehicle, $n = 8$ mice per group, and (R) IFNγ levels, $n = 5$ vehicle, $n = 6$ mice per group, were measured using ELISA, one-way ANOVA and Bonferroni's post-test were used, (P values: P: 0.006; 0.006; Q: 0.01; R: 0.03). Data information: (C–J, O–R) Data are mean ± SEM, and (K–M) median ± IQR, *$P < 0.05$, **$P < 0.01$, ***$P < 0.001$, ns non-significant. Source data are available online for this figure.

## Discussion

Our recent study described the pro-viral effects of CXCL4 on DENV replication. Elevated levels of this platelet chemokine in plasma correlated with high viremia in patients with febrile illness. The study also described a rescue effect of AMG487, a CXCR3 antagonist, on DENV2 replication in vitro (Ojha et al, 2019). Although AMG487 has anti-viral potential against DENV2, the therapeutic usage of this drug raised major concern following its withdrawal from the Phase II clinical trial against rheumatoid arthritis due to limited efficacy (Wijtmans et al, 2008). We have developed an alternate CXCR3 antagonist 7D, a small molecule that potentially inhibits DENV2 replication and improves mice survival. Importantly, 7D supplementation increases the synthesis of IFNs and neutralizing antibodies in DENV2-infected mice.

Compound 7D, having thermodynamically stable binding interaction with CXCR3, inhibited CXCL4-mediated DENV2 replication in CXCR3$^{+/+}$, but not in CXCR3$^{-/-}$ monocytes in vitro, suggesting CXCL4:CXCR3 axis as the crucial node of this signaling pathway. 7D also suppressed the replication of other serotypes, including DENV1, DENV3 and DENV4 in vitro, indicating a broad range of inhibitory effects against dengue infection. The supplementation with 7D at a concentration (8 mg/kg body weight) for 4 days rescued the dengue symptoms like thrombocytopenia, leukopenia and vascular leakage, as well as improved survival in AG129 mice. The cytokines, including TNFα (Malavige and Ogg, 2013; Yu et al, 2022) and IL1β (Kurane et al, 1991), which are known to activate vascular-endothelial cells and promote vascular leakage, were elevated in DENV2-infected mice. The vascular leakage phenotype was rescued along with reduced TNFα and IL1β levels in plasma of these infected mice after 7D treatment. Besides, 7D treatment also increased the synthesis of anti-viral cytokines IFNα/β and IFNλ in mice via CXCL4:CXCR3:p38:IRF3 signaling. The CXCL4-mediated activation of the CXCR3:p38 axis and its reversal by CXCL4-neutralizing antibodies have been described (Wang et al, 2019). Our previous study also described the CXCL4-mediated activation of CXCR3:p38 pathway and its reversal by either a CXCL4-neutralizing antibody or CXCR3-antagonist AMG487 (Ojha et al, 2019). Our current observations highlight the stimulatory role of 7D on IFN axis via similar pathway. Although the elevated IFNα/β do not explain the anti-viral response of these cytokines in IFNα/β/γ receptor-deficient AG129 mice, but the

elevated type-III interferon (IFNλ) in macrophages in these mice extends a protection against DENV2 infection upon 7D treatment. Our study also described the 7D-mediated elevation of IFNα/β levels in WT mice, protecting against DENV2 infection. Secretion of IFN by infected cells is an important strategy of primary defense mechanism against virus entry to the neighboring cells (Baron and Dianzani, 1994). Our study highlights the safe usage of 7D in promoting IFNα/β/λ synthesis against DENV infection. 7D is non-toxic to liver, kidney and blood cells, and has a plasma half-life of ~2.85 h in mice.

Another unique property of this compound is to stimulate the synthesis of antigen-specific antibodies in DENV2-infected mice. This phenomenon was not observed with AMG487 treatment. The elevated level of DENV2-neutralizing antibodies in peripheral blood directly correlated with a higher percentage of plasma cells in the spleen of the infected mice after 7D supplementation. Mechanistically, a higher expression of acetylated as well as phosphorylated STAT3 in plasma cells was observed in 7D-treated mice. Having the inhibitory effects on deacetylase Sirt-1 (Purushotham et al, 2022), 7D treatment might have promoted the acetylation of STAT3$^{K685}$, in turn phosphorylation of STAT3$^{Y705}$. The STAT3 acetylation and phosphorylation are known to play a crucial role in lymphocyte proliferation and differentiation (Limagne et al, 2017; Mackie et al, 2023). This suggests a stimulatory role of 7D:Sirt1:STAT3$^{K685}$:STAT3$^{Y705}$ axis in plasma-blast proliferation and generation of neutralizing antibodies against DENV2 in mice after 7D treatment. A STAT3 inhibitor decrease the effect of 7D in DENV2-infected mice, confirming the involvement of this axis. A similar stimulatory role of 7D:Sirt1:STAT3 signaling on antibody axis was observed in CXCR3$^{-/-}$ mice, that provided protection against dengue infection to these mice, lacking a functional 7D:CXCR3:IFN axis. 7D supplementation promoted the acetylation and phosphorylation of STAT3 in plasmablasts, in turn increasing the synthesis of neutralizing IgM and IgG against DENV2.

Together, our study identifies compound 7D as a stimulator of IFNα/β/λ synthesis via CXCL4:CXCR3:p38:IRF3 signaling, and also a booster for neutralizing-antibody generation by promoting Sirt1-mediated acetylation of STAT3 in plasmablasts; the molecule is thus capable of inhibiting dengue infection. The effects of 7D are outlined in Fig. 7.

Caveats in our study include the effects of 7D on other chemokines such as CXCL9/CXCL10 that also bind to the CXCR3

receptor and are implicated in DENV infection (Rothman, 2011). Also, a clear mechanism of 7D regulating type-II interferon IFNγ in DENV infection remains elusive.

# Methods

**Reagents and tools table**

| Reagent/resource | Reference or source | Identifier or catalog number |
|---|---|---|
| **Experimental models** | | |
| Vero E6 | ATCC | CRL-1586 |
| U937-DC-SIGN | ATCC | CRL-3253 |
| Mouse embryonic fibroblasts (WT and SIRT1-/-) | Isolated from WT and SIRT1-/- mice embryos. (Kolthur-Seetharam et al, 2006) | A kind gift from Dr Ullas S Kolthur, CDFD, Hyderabad, India |
| L929 cells | ATCC | CCL-1 |
| C636 | ATCC | CRL-1660 |
| MEG01 | ATCC | CRL-2021 |
| AG129 mouse (IFN α/β/γ R-/-) | Marshall BioResources | |
| C57BL/6J | The Jackson Laboratory | JAX stock #000664 |
| B6.129P2-Cxcr3tm1Dgen/J mouse | The Jackson Laboratory | JAX stock #005796 |
| **Antibodies** | | |
| 4G2 | Merck-Millipore | MAB10216-I |
| IFN-λ 2/3 | R&D | MAB17892 |
| p53 | Cloud clone | PAA928Hu01 |
| P-p38 | Cell Signalling Technology | 4511s |
| p38 | Cell Signalling Technology | 9212s |
| IRF3 | Cell Signalling Technology | 4302S |
| P-IRF3 | Cell Signalling Technology | 29047S |
| Actyl-p53 | Cell Signalling Technology | 2525s |
| P-STAT3(Y705) | Cell Signalling Technology | 4113S |
| Actyl-STAT3(K685) | Cell Signalling Technology | 2523S |
| STAT3 | Cell Signalling Technology | 4904S |
| SIRT1 | Cell Signalling Technology | 8469S |
| β-Actin | Cell Signalling Technology | 3700S |
| dsRNA | Cell Signalling Technology | 76651L |
| CD45-APC (1:100) | BioLegend | 109814 |
| CD19-BV421 (1:100) | BioLegend | 115538 |
| CD3-FITC (1:100) | BioLegend | 100204 |
| CD138-PE/CY7 (1:150) | BioLegend | 142514 |
| B220-PE (1:150) | BioLegend | 103208 |
| LY6c-BV421 (1:75) | BioLegend | 128032 |

| Reagent/resource | Reference or source | Identifier or catalog number |
|---|---|---|
| GL7-Unconjugated (1:200) | BioLegend | 144602 |
| LY6g-FITC (1:100) | Invitrogen | 11-9668-82 |
| CD11b-PerCP/CY5.5 (1:75) | Invitrogen | 45-0112-82 |
| CD41a-PE (1:100) | Invitrogen | 12-0411-82 |
| DENV NS1 | Invitrogen | PA5-32207 |
| Alexa flour 488 anti-mouse IgG | Invitrogen | A28175 |
| Dylight 650 anti-rat IgM | Invitrogen | SA5-10013 |
| Biotinylated anti-mouse IgG | Southern Biotech | 6170-08 |
| Biotinylated anti-mouse IgM | Southern Biotech | 1021-08 |
| HRP conjugated anti-mouse IgG | Immunotag | 786-R38 |
| HRP conjugated anti-rabbit IgG | Immunotag | 786-R39 |
| **Oligonucleotides and other sequence-based reagents** | | |
| RT-qPCR primers and Genotyping primers | | Appendix Table S2 |
| Sequencing Primers | | Appendix Table S2 |
| **Chemicals, enzymes and other reagents** | | |
| RPMI media | Sigma | R4130-1L |
| DMEM media | Gibco | 12800-017 |
| FBS | Gibco | 10270-106 |
| Pen/strep | Sigma | P4333 |
| Leibovitz's L15 media | Himedia | AL011S |
| Amicon filters | Merck-Millipore | UFC9100 |
| Triton X100 | Sigma | T8787 |
| BSA | BioString | BS100053 |
| CD61 micro beads | Miltenyi Biotec | 130-109-678 |
| Mice IFN-λ 2/3 ELISA | R&D | DY1789B-05 |
| DAPI | CST | 4083S |
| Prolong gold | CST | 9071S |
| cDNA kit | Bio-Rad | 1708891 |
| SYBR green master mix | Bio-Rad | 1725124 |
| RIPA lysis buffer | Sigma | R0278 |
| Protease and phosphatase inhibitor | Thermo | A32959 |
| Stattic | Target mol | T6308 |

| Reagent/resource | Reference or source | Identifier or catalog number |
|---|---|---|
| AMG487 | Medchem Express | HY-15319 |
| IL-1β CBA | BD Biosciences | 560232 |
| TNF CBA | BD Biosciences | 558299 |
| IL-6 CBA | BD Biosciences | 558301 |
| Evan's blue dye | Sigma | E2129 |
| DENV2 antigen | Microbix | EL-22-02 |
| TMB chromogen substrate | Thermo | 34022 |
| Stop solution | Thermo | N600 |
| Mice IFNα ELISA | Elabsciences | E-EL-M3054 |
| Mice IFNβ ELISA | Elabsciences | E-EL-M0033 |
| Mice PF4 ELISA | R&D | MCX400 |
| Human PF4 ELISA | Elabsciences | E-EL-H6184 |
| AEC substrate | Vector Labs | SK-4200 |
| MTT | Himedia | RM1131 |
| DMSO | Sigma | D8418 |
| QIAamp viral RNA mini kit | Qiagen | 52904 |
| **Software** | | |
| Schrödinger suite | Schrödinger | |
| AMBER22 | https://ambermd.org/AmberMD.php | |
| ImageJ Fiji | https://imagej.net/ij/download.html | |
| CBA analysis software | BD Biosciences | |
| FlowJo | BD Biosciences | |
| Biorender | https://www.biorender.com/ | |
| GraphPad Prism 8.0 | GraphPad | |
| **Other** | | |
| Leica Confocal DMI 6000 TCS-SP8 microscopes. | Leica | |
| Quant StudioTM 6 Flex Real-Time PCR System | Applied Biosystems (Thermo Scientific) | |
| BD FACS-Verse | BD Biosciences | |
| BD FACS-aria | BD Biosciences | |
| Spectra max i3x multi-mode reader | Molecular devices | |
| S6 Universal M2 | Immunospot | |
| Illumina Miseq | Illumina | |
| Agilent 1200 Infinity | Agilent | |

## In silico screening of CXCR3 antagonists

**Modelling of CXCR3 and system preparation**: In the absence of the crystal structure of CXCR3, the 3D model of CXCR3 was constructed using MODELLER (Petrovskiy et al, 2023). BLAST (from NCBI) was analysis was performed to identify the suitable template with considerate identity and similarity. The BLAST provided the template PDB ID: 4MBS (resolution 2.71 Å) (Tan et al, 2013) of CCR5 which was used to model CXCR3 (Appendix Fig. S1A). The model was validated for Ramachandran plot using Maestro (Appendix Fig. S1B). The crystal structure of CCR5 bound with MRV was further used as reference to compare the outcomes of CXCR3. Hence, both the structures were prepared using the Protein Preparation Wizard module of Maestro (Schrödinger Release 2022-1) (Sastry et al, 2013; Srivastava et al, 2018). In the preparation process, the hydrogen and bond orders were added using PRIME. The hydrogen bond (HB) optimization and restrained minimization was also done for the systems using OPLS3 force field model (Roos et al, 2019).

**Compound library retrieval and preparation**: Using the CXCR3 structure we identified novel hits after careening (virtually) from 1.1 million molecules (curated in-house for virtual screening). The ligands of this library were prepared using the LigPrep module of Maestro with Epik to add their protonation and ionization states (Schrodinger Suite 2020-1). The force field was set the same as with the receptor in this process (Kumari et al, 2021). The compounds were prepared at physiological pH conditions, desalted, tautomer's were generated and finally the compounds were minimized. The grid was created around residues of MRV as the CXCR3 and CCR5 are homologs of each other, and it was hypothesized that similar binding site could be used for identifying hits against CXCR3.

**Molecular docking and virtual screening**: The virtual screening workflow (VSW) was implemented by using the Glide module in Maestro (Halgren et al, 2004). A grid in CXCR3 was generated at similar location of CCR5 where MRV is bound as they are homologs of each other, and it was hypothesized that similar binding site could be used for identifying hits against CXCR3. The molecules were filtered out using Lipinski's rule of five and other physiochemical properties relevant for small molecules by QikProp (Mittal et al, 2021a), and finally the reactive functional groups were removed by LigFilter. The virtual library compounds was screened on the targeted site. VSW has three docking steps. The first stage is HTVS (high throughput screening), where we screened 60% of the top compounds to have all good scoring states (ionization or tautomeric states), followed by Glide SP (standard precision) docking (50%) and Glide XP (extra precision) docking (40%) stages. After the docking, the binding conformations were studied to elucidate the essential interactions between protein and ligand.

**Prime MM-GBSA calculations**: Post-docking minimizations have also been used to improve the geometry of the docking structures. The final poses from the Glide docking were post-processed by using the Prime MM-GBSA module. The results of the docking were then quantified on the consensus of docking scores and Prime MM-GBSA energy. Prime MM-GBSA uses a continuum solvation model for refinement. The solvation model VSGB2.1 (variable-dielectric generalized born model) was used here, which incorporates residue-dependent effects and water is the solvent. OPLS3e force field was applied and the minimize-sampling method was used to minimize all atoms in each residue (Roos et al, 2019).

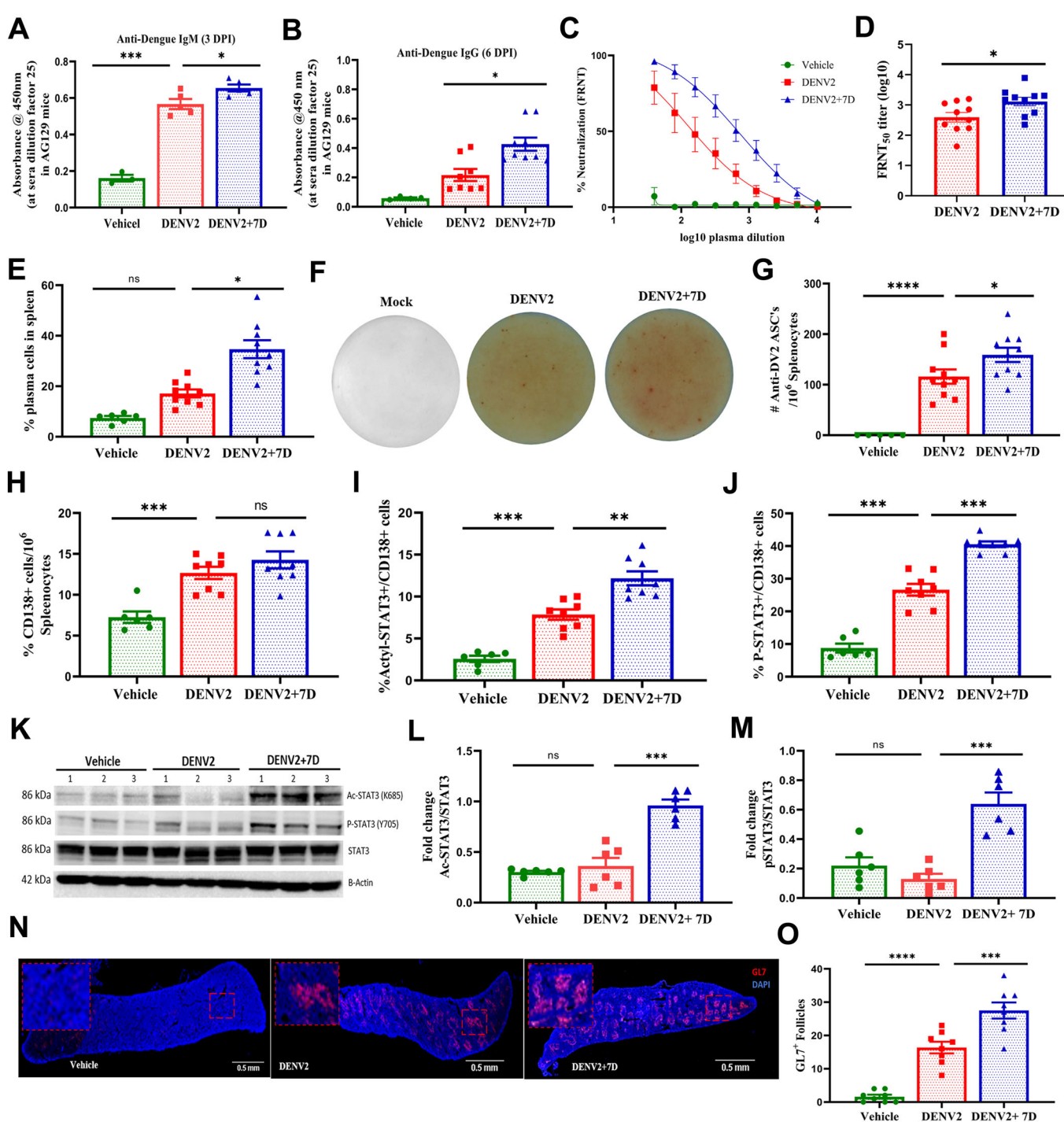

The MM-GBSA method was used to calculate the relative binding affinity of ligands to the receptor (in kcal/mol). Because the MM-GBSA binding energies are estimates of binding free energies ($\Delta$Gbind), a more negative value implies a better binding. Prime MM-GBSA calculates the energy of optimized free receptors, free ligand, and a complex of the ligand with a receptor.

**ADME properties**: A set of ADME-related properties for the control and library molecules were predicted by QikProp (Mittal et al, 2021b). The ADME descriptors predicted for the current

study are molecular weight (MW), solvent accessible surface area (SASA), number of hydrogen bond donors (donorHB) and acceptors (accptHB), number of rotatable bonds (rotor), predicted octanol/water partition coefficient (QPlogPo/w), predicted aqueous solubility (QPlogS), and percent human oral absorption (Mittal et al, 2021a).

**Molecular dynamics (MD) simulation to assess the stability of identified hits**: AMBER19SB force field and OPC water model were used for assessing the stability of hit 7D-CXCR3, AMG487-CXCR3

Figure 5. 7D treatment enhances DENV2-specific antibodies in infected mice.

(A–D) DENV2-specific antibodies (A) IgM at 3 DPI, and (B) IgG at 6 DPI were measured in mice plasma from the above experiments (Fig. 4A) using ELISA. A: $n = 3$ vehicle, $n = 5$ mice per group, one-way ANOVA and Bonferroni's post-test, B: $n = 4$ vehicle, $n = 8$ DENV2, $n = 9$ DENV2 + 7D, Mann–Whitney $U$ test was used, ($P$ values: A: 0.001; 0.03; B: 0.01). (C) FRNT$_{50}$ curves of neutralization activity of mice serum on DENV2 propagation in Vero cells, $n = 3$ vehicle, $n = 10$ mice per group. (D) Graph of the above values, $n = 10$ mice per group, Mann–Whitney $U$ test was used ($P$ value: 0.02). Sera dilution assay is mentioned in Appendix Fig. S5A,B. (E) Increased plasma cells (CD138$^{+ve}$B220$^{lo/-ve}$ CD19$^{+ve}$) percentage in 7D-treated mice spleen was measured using flow cytometry, $n = 6$ vehicle, $n = 9$ mice group, Kruskal–Wallis test is used ($P$ value: 0.02). (F, G) Splenocytes from the above mice were used to quantitate DENV2-specific antibody secreting cells using ELISPOT assay, (F) representative image and (G) graph of the above values, $n = 5$ vehicle, $n = 10$ mice per group, one-way ANOVA and Bonferroni's post-test were used ($P$ values: 0.0001; 0.04). (H–J) Intracellular levels of (H, I) Acetylated (Ac)-STAT3 and (H, J) phosphorylated (P)-STAT3 in CD138$^{+ve}$ cells was assessed by flow cytometry, $n = 6$ vehicle, $n = 8$ mice per group, one-way ANOVA and Bonferroni's post-test were used ($P$ values: H: 0.0009; I: 0.001; 0.005; J: 0.001; 0.001). Gating strategy is mentioned in Appendix Fig. S4D. (K–M) Western blot analysis of (K, L) Ac-STAT3 and (K, M) P-STAT3 from the splenocytes normalized to β-actin. Densitometry of the above blots, $n = 6$ mice per group, one-way ANOVA and Bonferroni's post-test were used ($P$ values: L: 0.001; M: 0.001). (N, O) Germinal center in spleen. (N) Immunofluorescent images of GL7-expressing follicles in spleen sections, GL7 (red) and DAPI (blue) staining. (O) Increased GL7-expressing follicles in 7D-treated mice, $n = 8$ mice per group, one-way ANOVA and Bonferroni's post-test were used ($P$ values: 0.0001; 0.0002). Data information: (A–C, E, G–J, L, M, O) Data are mean ± SEM, and (D) median ± IQR, *$P < 0.05$, **$P < 0.01$, ***$P < 0.001$, ****$P < 0.0001$, ns non-significant. Data from similar experiment in WT mice is described in Fig. EV4. Source data are available online for this figure.

and further comparing it with the crystal of MRV-CCR5 respectively. State-of-the-art all-atom MD simulations were carried out with the AMBER22 package for the model proteins (Case et al, 2022). The solute was placed within a cubic box ensuring a minimum distance of 12.0 Å between any protein atom and the edge of the box filled with explicit water molecules and counterions. Briefly, geometry optimizations were carried out with a two-step protocol: (i) up to 10,000 cycles (2000 of steepest descent plus 8000 of conjugate gradient) with harmonic restraint (k = 1 kcal mol$^{-1}$ Å$^{-2}$) on non-hydrogen atoms of the solute; (ii) up to 10,000 conjugate gradient cycles with no restraints. Next, heating up to 310 °K was achieved by linearly increasing the temperature within 100 ps of NVT MD, while imposing restraints of 1 kcal mol$^{-1}$ Å$^{-2}$ on non-hydrogen atoms of solute. Restraints were then released for 100 ps and, as a last step preceding the productive dynamics, 1 ns of NPT MD was carried out to relax the simulation box. Finally, an MD simulation of 100 ns duration for protein in explicit water solution under the NPT ensemble was performed. Temperature and pressure were regulated at 310 °K and 1.013 bar using a Langevin thermostat (damping constant 5 ps-1) and the Nosé–Hoover–Langevin piston pressure control (Di Pierro et al, 2015). Electrostatic interactions were evaluated using Soft Particle Mesh Ewald schemes with 1.0 Å grid spacing and a cut-off of 12.0 Å, i.e. the same used for Lennard-Jones interactions.

## Cell culture and viruses

Vero E6 (CRL-1586, ATCC, USA) cells were cultured in Dulbecco's modified eagle's medium (DMEM; Gibco, USA) supplemented with 10% fetal bovine serum (FBS, Gibco) and 1% penicillin/streptomycin (Sigma-Aldrich, USA). U937-DC-SIGN cells (CRL-3253, ATCC), MEG-01 cells (CRL-2021, ATCC), human and mouse monocytes were cultured in RPMI (Sigma-Aldrich) supplemented with 10% FBS and 1% penicillin/streptomycin. C6/36 cells (CRL-1660, ATCC) were cultured in Leibovitz's L15 media (Hi-media, India) supplemented with 10% FBS and 1% penicillin/streptomycin. Wild-type and SIRT1$^{-/-}$ Mouse Embryonic Fibroblasts (MEF's, a kind gift from Dr Ullas S Kolthur, Centre for DNA Fingerprinting & Diagnostics, Hyderabad, India; Kolthur-Seetharam et al, 2006) were cultured in high glucose Dulbecco's modified eagle's medium (DMEM) supplemented with 10% FBS and 1% penicillin/streptomycin. The L929 cells (CCL-1, ATCC) are cultured in high glucose Dulbecco's modified eagle's medium (DMEM) supplemented with

10% FBS and 1% penicillin/streptomycin. After seven days, cell supernatant was collected and used as conditioned-media for macrophage differentiation from bone marrow cells.

DENV2 isolate P23085 INDI-60 (GenBank Accession No. KJ918750.1), DENV2 (Accession No: EU081177.1; a kind gift from Dr Ashley L. St. John, NUS, Singapore), DENV4 (strain H241), DENV1 (EU858545) and DENV3 (strain H87) were used for in vitro or in vivo infection. All virus strains were propagated in C6/36 cells. Briefly, 70% confluent C6/36 cells were infected with MOI ~0.1 of DENV2 strains for 2 h in L15 media supplemented with 2% FBS. After 2 h, infection was removed and cells were washed with PBS and fresh media with 10% FBS and 1% penicillin/streptomycin. Cells were incubated at 28 °C for 5 days. Cell supernatant was harvested and concentrated (10-fold) using Amicon filter (Merck Millipore). For virus quantification, focus forming assay was carried out to determine the virus titer for all virus strains and serotypes. Vero E6 cells were seeded at a density of $0.075 \times 10^5$ cells/well in a 24-well plate and cells were allowed to reach 80% confluency. At 80% confluency 200 µl of serially diluted virus was added to the wells and incubated. After 2 h, virus was removed, fresh media with 2% FBS was added and incubated for 3 days at 37 °C in 5% $CO_2$. Following cells were fixed with 2% PFA and the immunostaining for virus detection was performed using mAb 4G2 (pan-DENV anti-envelope antibody, MAB10216-I, Merck-Millipore), followed by secondary antibody, goat anti-mouse IgG, AF488 conjugated (A28175, Invitrogen) prepared in permeabilization buffer (0.1% Triton x100 with 2% BSA in PBS). The foci were visualized and counted using fluorescence microscopy. All cell lines were authenticated before use and monitored for mycoplasma contamination and the experiments were performed in mycoplasma free cells.

## Isolation and primary culture of mice megakaryocytes and BMDM's

Hind limbs of WT mice were collected, and bone marrow was flushed out using a syringe needle. The bone marrow is dissociated by pipetting and passed through a 70-µm cell strainer and resulting single cell suspension is used to isolate the megakaryocytes and culture bone marrow derived macrophages (BMDM's). Megakaryocytes from the bone marrow cells were isolated by direct positive selection method using CD61 micro beads and large cell separation columns from Miltenyi Biotec, Germany, as per the manufacturer's

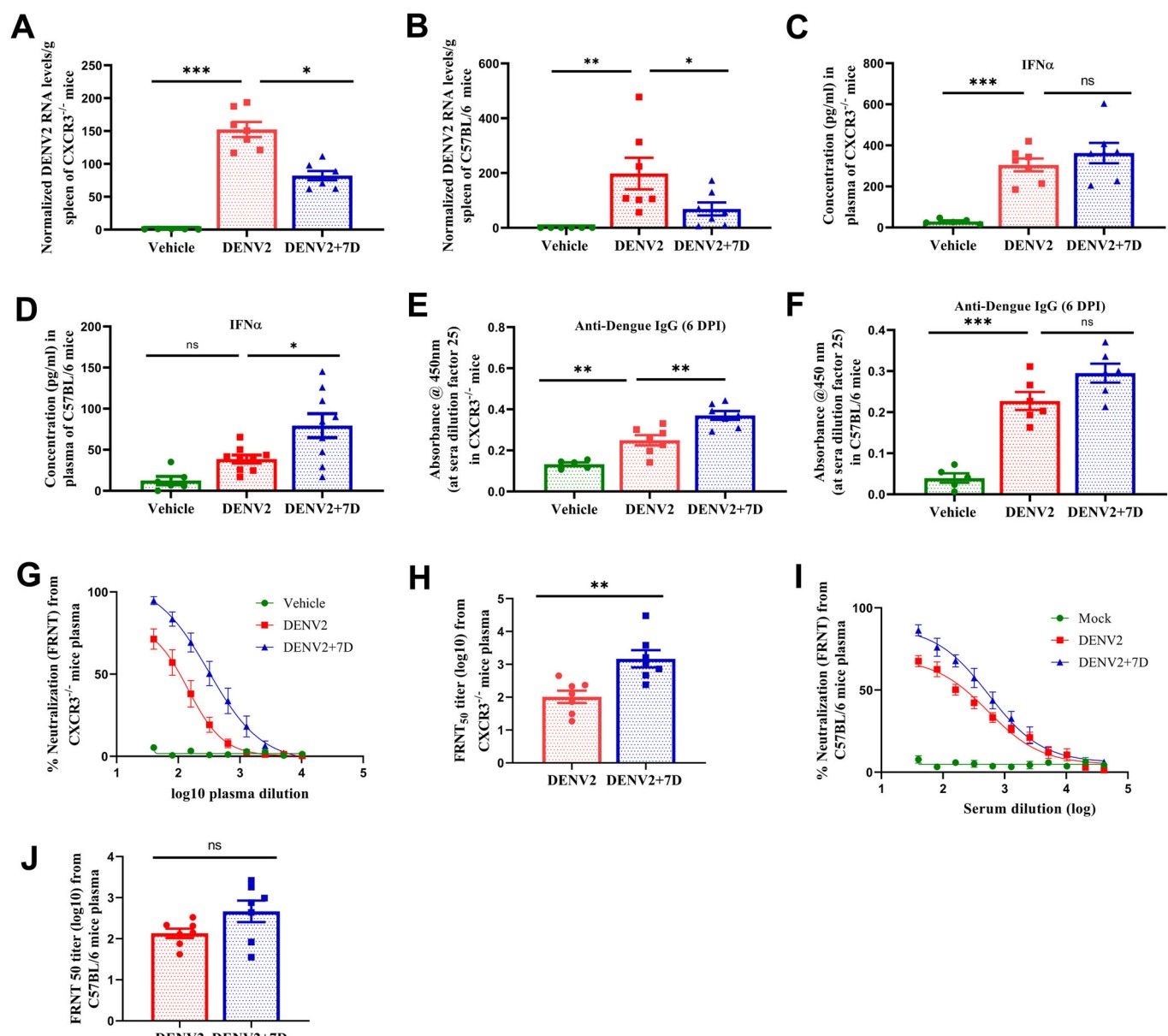

**Figure 6. 7D treatment reduces DENV2 load by enhancing neutralizing antibodies in CXCR3−/− and WT mice.**

As mentioned in the above Fig. 4A, a similar experiment was performed in CXCR3−/− and WT C57BL/6 mice at 6 DPI. (A, B) Spleen was isolated and processed for measuring dengue viral genome copies were quantified in (A) CXCR3−/−, n = 5 vehicle, n = 7 mice per group, one-way ANOVA and Kruskal–Wallis test, (P values: 0.0002; 0.03), and (B) WT, n = 6 vehicle, n = 7 mice per group, one-way ANOVA and Bonferroni's post-test, (P values: 0.004; 0.04) using qRT-PCR. (C, D) IFNα levels were measured in plasma of these mice (C) CXCR3−/−, n = 5 vehicle, n = 7 mice per group, and (D) WT, n = 6 vehicle, n = 9 mice per group, one-way ANOVA and Bonferroni's post-test were used for both cases, (P values: C: 0.0003, D: 0.01). (E, F) DENV2-specific IgG levels were measured in mice plasma of (E) CXCR3−/−, n = 5 vehicle, n = 7 mice per group, and (F) WT, n = 5 vehicle, n = 6 mice per group, using ELISA. One-way ANOVA and Bonferroni's post-test were used for both cases, (P values: E: 0.003; 0.0013, F: 0.001). (G–J) FRNT50 curves and graph of neutralization activity of mice plasma was measured in (G, H) CXCR3−/−, n = 3 vehicle, n = 7 mice per group, and (I, J) WT, n = 6 vehicle, n = 7 mice per group. Mann–Whitney U test was used for analysis for both cases, (P value: 0.003). Sera dilution assay is mentioned in Appendix Fig. S5. Data information: (A–F, G, I) Data are mean ± SEM, and (H, J) median ± IQR, *P < 0.05, **P < 0.01, ns non-significant. Source data are available online for this figure.

protocol. The bone marrow cells were cultured in the presence of 10% L929-conditioned media for 5 days to differentiate into macrophages. BMDM from AG129 mice were used for measuring type-III interferon (IL-28A/B or IFN-λ 2/3) using ELISA kit from the R&D Systems, USA. A blocking antibody against mouse IFN-λ 2/3 (MAB17892, R&D) was procured and used to block the function of this interferon in vitro.

## Immunofluorescence staining

U937-DC-SIGN cells or monocytes were fixed with 4% PFA and blocked with 5% BSA prepared in PBS with 0.1% Triton X100. Cells were stained with dsRNA (76651L, Cell Signaling Technology, USA) or NS1 (PA5-32207, Invitrogen) and related secondary antibodies. Cells were stained with DAPI (Cell Signaling

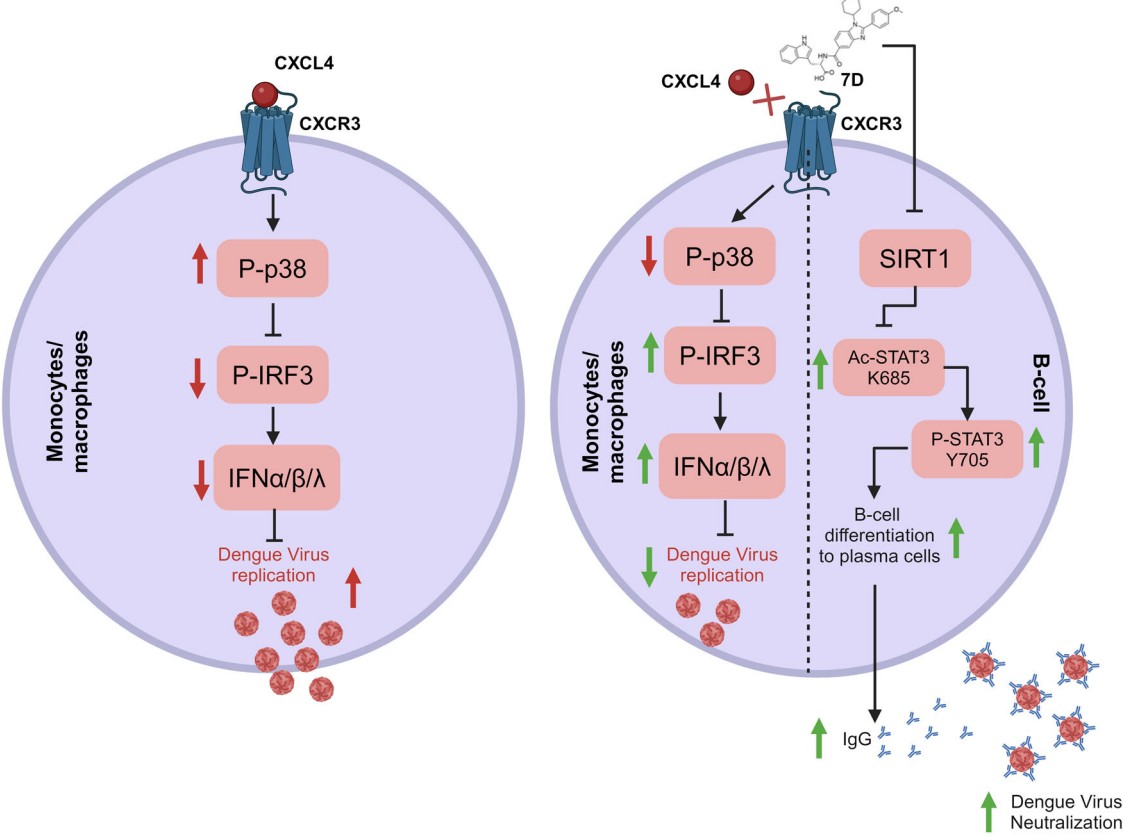

**Figure 7. Schematic describes the CXCL4-mediated activation of CXCR3:p38:IRF3 signaling, in turn suppression of IRF3 and IFNα/β/λ in monocytes/macrophages.**

Conversely, 7D supplementation reverses the above signaling and improves IFNα/β/λ synthesis. Besides, 7D increases acetylation and phosphorylation of STAT3, in turn promotes plasmablast proliferation and IgG synthesis via suppression of deacetylase activity of Sirt-1.

Technology) and were mounted on to cover slips using prolong gold anti-fade reagent. Images were captured using a Leica Confocal DMI 6000 TCS-SP8 microscope (Leica Microsystems, Germany) at 63x oil immersion objective (NA 1.4) Plan Apo objectives and quantified. Imaging was performed using Z-stacks at 0.25 μm per slice by sequential scanning and ImageJ Fiji software was used to generate cross-sectional and maximum intensity projection images. Background noise was measured and deducted from the total fluorescence intensity of the cells and corrected cell maximum fluorescence intensity was plotted from at least 40 cells from three independent experiments. Mouse spleen tissue were processed to frozen tissue sections and permeabilized with 0.5% TritonX-100 in PBS. The sections were blocked with 5% goat serum. The slides were stained with anti-rat GL7 primary antibody (144602, BioLegend) at 4 °C overnight followed by Dylight650-conjugated secondary antibody (SA5-10013, Invitrogen). Spleens were stained with DAPI (Cell Signaling Technology) and were mounted on to cover slips using prolong gold anti-fade reagent. Images were captured using a Leica Confocal DMI 6000 TCS-SP8 microscope (Leica Microsystems, Germany) at 10× objective Plan Apo objectives and quantified. Imaging was performed using Z-stacks at 1 μm per slice by sequential scanning and ImageJ Fiji software was used to generate cross-sectional and maximum intensity projection images.

## Quantitative real time PCR

Total RNA from the mice tissues or cell pellets was extracted using RNAiso (Takara Bio, Japan) followed by phenol-chloroform treatment. The first strand cDNA was synthesized from 1 μg RNA using cDNA synthesis kit (BioRad, USA) as per manufacturer's protocol. The cDNA was used for real-time PCR using SYBR green supermix (BioRad) in an Applied Biosystems Quant StudioTM 6 Flex Real-Time PCR System. The primers set used for detection of the respective genes are tabulated in Appendix Table S2.

## Western blotting

Cells or mice monocytes were lysed using RIPA lysis buffer (Sigma-Aldrich) with Protease-phosphatase inhibitor (Thermo Scientific). The protein on the SDS-PAGE gel were transferred on to PVDF membrane followed by immunoblotting using the primary antibodies against, DENV NS1 (PA5-32207, Invitrogen), P-p38 (4511s), p38 (9212s), IRF3 (4302s), P-IRF3 (29047s), Actyl-p53 (2525s), P-STAT3(Y705) (4113s), Acetyl-STAT3(K685) (2523s), STAT3 (4904s), SIRT1 (8469s) and β-Actin (3700s) from Cell Signaling Technology, p53 (PAA928Hu01, Cloud clone, USA). Secondary HRP conjugated anti-mouse (786-R38) and anti-rabbit

(786-R39)cs IgG antibodies (ImmunoTag, USA) were used as required to develop the blots.

## DENV2 adaptation in mice

The DENV2 (P23085 INDI-60) was amplified in C6/36 cells maintained in L15 medium and concentrated using 100 kDa, Amicon filters and stored in −80 °C. DENV2 ($10^7$ FFU) was injected in 6–8 weeks old AG129 mice intravenously. Mice were anesthetized by ketamine/xylazine, and blood was collected via cardiac puncture followed by harvesting of serum at day 3 post-infection. The serum was overlaid on to the monolayer of C6/36 cells. After 2 h, fresh media was added and incubated at 28 °C for 5–6 days to amplify the virus. After 6 days, cell supernatant was collected and concentrated (tenfold) using Amicon filters (100 kDa). Virus (~$10^5$–$10^6$ FFU) collected from C6/36 cells, was injected intravenously in AG129 mice, described in Appendix Fig. S2A. This entire procedure was repeated 8 times to obtain mouse-adapted DENV2 (P8-P23085 INDI-60). The symptoms of mice infected with $10^5$ FFU of passage 8 adapted DENV2 were observed like weight loss, hunch back posture and ruffled fur. Most of the mice died between 7 and 8 DPI. The mouse models of DENV were developed as mentioned (Sarathy et al, 2015; Tan et al, 2010).

## Mouse infection

The AG129 (IFNα/β/γR$^{-/-}$ 129/Sv), CXCR3$^{-/-}$ in BL6 background and C57BL/6 mice were obtained from the Jackson Laboratory, USA and housed in individually tagged well-ventilated cages at a temperature of 22 ± 1 °C, relative humidity of 55 ± 10%, and a light/dark cycle of 12 h/12 h, with free access to food and water at experimental animal facility (EAF), RCB, Faridabad, India. All experiments with AG129 mice were performed in 6–8 weeks old mice of either sex. The mice were infected intraperitonially (i.p.) with 100 μl viral suspension containing $1 \times 10^5$ FFU mouse-adapted DENV2 (P8-P23085 INDI-60). The mice were injected with either compound 7D i.p. (8 mg/kg body weight/ day) or vehicle (PBS) for 4 days post-infection. Besides, CXCR3$^{-/-}$ and C57BL/6 WT mice of 4–5 weeks were infected (i.p.) with 100 μl viral suspension of $1 \times 10^6$ FFU DENV2 (EU081177.1). 7D was administered as mentioned above. An inhibitor to STAT3, Stattic was purchased from the TargetMoI Chemicals, Boston, USA, and used for in vivo study in mice at a concentration (10 mg/kg body weight/day for 4 days). The blinding protocol was not used for animal experiments.

## Cytokine measurement by cytometric bead array (CBA)

The CBA array was performed to measure cytokines such as IL-1β, TNF-α, and IL-6 from plasma samples collected from dengue infected mice of different treatment groups as described in results and analyzed by CBA analysis software (BD Biosciences) (Ojha et al, 2019).

## Quantitation of vascular permeability

Vascular permeability in infected mice was evaluated through leakage of Evan's blue dye from mice blood vessels. The treatment was continued until 4-day post-infection (DPI) as previously described and at day 5, mice were administered 200 μL of 0.5%

Evan's blue dye intravenously in tail and dye was allowed to circulate for 2 h. After that mice were anesthetized and perfused extensively with PBS through heart. Tissues such as, liver, spleen, heart, brain, small intestine, lung, and kidney were collected, weighed and Evan's blue dye was extracted from tissues by incubation in 1 mL formamide at 37 °C for 24 h. The tissues debris was settled by centrifugation at 5000 rpm for 15 min then after 100 μL of supernatant was collected and the concentration of Evan's blue dye was quantified at absorbance of 610 nm. The results were demonstrated as optical density ($OD_{610}$) per gram of tissue mass as described (St John et al, 2013).

## Flow cytometry

Mice PBMCs or single cell suspension of spleenocytes were resuspended in FACS buffer and incubated with $F_C$ block at room temperature for 15 min. In total, $10^6$ cells were stained with antibodies CD45 (109814, BioLegend), LY6g (11-9668-82, Invitrogen), CD3 (100204, BioLegend), CD19 (115538, BioLegend), CD138 (142514, BioLegend) and B220 (103208, BioLegend), CD11b (45-0112-82, Invitrogen), LY6c (128032, BioLegend), CD41 (12-0411-82, Invitrogen) in 50 μl FACS buffer. For intercellular staining, cells were fixed with fixation buffer (Invitrogen) and permeabilized using 1× permeabilizing buffer (Invitrogen) after extracellular staining. Antibodies staining intercellular P-STAT3$^{Y705}$ (4113s, Cell Signalling Technology) and Acetyl-STAT3$^{K685}$ (2523s, Cell Signaling Technology) were resuspended in permeabilizing buffer and incubated with cells for 1 h and incubated with secondary antibody conjugated with alexa-fluor 488 (A28175, Invitrogen) for 1 h. Cells were washed and acquired using BD FACS-Verse and BD FACS-aria (BD Biosciences). Data was analyzed using FlowJo software (FlowJo LLC, Oregon) (Ojha et al, 2019).

## ELISA

Maxisorp 96-well microplates (Thermo Fisher) were coated with fixed DENV2 antigen (Microbix, Canada) at a concentration of 1 μg/ml in PBS overnight at 4 °C and washed with PBS with 0.1% Tween-20 (PBS-T). Plates were blocked with 2.5% w/v bovine serum albumin in PBS-T for 2 h at room temperature. Plasma, two-fold dilution, was incubated with the antigen for 1 h at room temperature, washed with PBS. Anti-mouse IgG HRP conjugated (786-R38, Immuno Tag) and biotinylated anti-mouse IgM (1021-08, SouthernBiotech) secondary antibodies were added and incubated at room temperature for 1 h and washed with PBS. Plates were washed with PBS-T and incubated with streptavidin-HRP (SouthernBiotech) at 1:5000 for 30 min. Plates were then developed with TMB chromogen solution (Thermo Fisher). Stop solution (Thermo Fisher) was added to stop the conversion of TMB and optical density was measured using spectra max i3x multi-mode reader (Molecular Devices, USA). Mice IFNα and IFNβ (Elabscience) and PF4 (R&D systems) were measured using ELISA as per the manufacturer instructions. Human PF4 ELISA (Elabscience, USA).

## ELISPOT

Elispot plates (Merck Millipore) were precoated with fixed DENV2 antigen (Microbix) at a concentration of 1 μg/ml in PBS overnight

at 4 °C and washed with PBS 3× and serially diluted splenocytes starting from $2 \times 10^5$ cells. Plates were incubated overnight at 37 °C with 5% $CO_2$. Next day cells were removed, and plates are washed with PBS-T before incubation with secondary biotinylated anti-mouse IgG (6170-08, SouthernBiotech) at 1:5000 for 2 h. Plates were washed with PBS-T and incubated with streptavidin-HRP (SouthernBiotech) at 1:5000 for 30 min. Plates were washed with PBS-T and spots were developed using AEC substrate (Vector Laboratories, USA).

## Neutralization assay

Neutralizing antibody titers were calculated by focus reduction neutralization test (FRNT) assay. Serial dilutions of each mice plasma were incubated with 50–60 FFU of DENV2 for 1 h at 37 °C. The virus-antibody immune complexes were transferred to the Vero E6 cell monolayer and plates were incubated for 2 h at 37 °C in 5% $CO_2$ incubator for virus adsorption. The DMEM media containing 10% FBS with 1% penicillin–streptomycin was overlaid on infected Vero cells monolayer and incubated for 3 days. After incubation cells were washed with PBS and fixed with 2% paraformaldehyde. The immunostaining for virus detection was performed using mAb 4G2 (pan-DENV anti-envelope antibody, MAB10216-I, Merck-Millipore), followed by secondary antibody, goat anti-mouse IgG, (AF488 conjugated, A28175, Invitrogen) prepared in permeabilization buffer (0.1% triton X100 with 2% BSA). The foci were visualized and counted using fluorescence microscope. The 50% neutralization titre $NT_{50}$ was calculated from non-linear regression curves in GraphPad Prism version 8.0 as mentioned (Whiteman et al, 2018).

## Cell viability and cytotoxicity assay

The U937-DC-SIGN cells were seeded in triplicates at density of ~15,000–20,000 cells per well in 96-well plates in 200 μL of RPMI (1640) medium containing 10% FBS, 1% penicillin–streptomycin and 2 mM L-glutamine and only 200 μL RPMI medium was kept as a negative control in other well. The drugs were added at two concentrations as mentioned in the Suppl. Fig. S1B, and drugs treated cells were incubated for 48 h at 37 °C in 5% $CO_2$ incubator. The cells without drugs treatment were used as a control. After 48 h, 10 μL of 5 mg/mL stock of MTT [3-(4,5-dimethylthiazol-2-yl)-2,5-diphenylte-trazolium bromide] was added. The plate was incubated for 3 h at 37 °C. MTT converts in Formazan crystals and this crystal was dissolved by adding the 50 μL of DMSO followed by incubation for 10 min at 37 °C. Absorbance taken at 570 nm using the ELISA reader as mentioned (Patil et al, 2018; Kumar et al, 2018).

## Virus whole genome sequencing

DENV2 viral RNA was extracted using QIAamp viral RNA mini kit (Qiagen, USA) as per the manufacturers protocol. cDNA was prepared from the RNA as a template using a cDNA specific primer with high-capacity cDNA reverse transcription kit (Thermo Fisher). Whole viral genome was amplified into six fragments from the cDNA using the primer pairs as mentioned in Appendix Table S2. PCR products were run on agarose gel and gel purified. Purified PCR products are further cleaved into short fragments and insertion of index 1 and index 2 were performed as per the manufacturers

protocol. Library were prepared after discarding the excess product and sample quantitation. The samples were sequenced using Illumina MIseq (Illumina, USA). Links and poor-quality reads were processed from the resulted raw data. The DENV2 genome sequence was then assembled from the reads using DENV2 (P23085 INDI-60, GenBank Accession No. KJ918750.1) as a reference sequence. The coverage of each nucleotide position on the gene sequence was calculated using SAM tools.

## HPLC analysis of 7D in plasma samples

The stock solution of compound 7D was prepared by dissolving the compound in dimethyl sulfoxide (DMSO) to achieve a final concentration of 20 mM. This stock solution was carefully stored under appropriate conditions to maintain its stability. To establish a standard curve for compound 7D, working solutions with concentrations ranging from 1 to 150 μM were made by diluting in methanol. Subsequently, 1 μL of each prepared sample was introduced into the High-Performance Liquid Chromatography (HPLC) system for analysis. Plasma samples collected at various time-points during the study were subjected to HPLC analysis to determine the concentration of compound 7D. Sample preparation involved measuring 100 μL of plasma sample for each timepoint and adding 2 mL of ethyl acetate. The resulting mixture was vigorously vortexed for exactly 1 min to ensure thorough mixing, leading to phase separation, with the ethyl acetate layer being collected. The ethyl acetate extract was then subjected to vacuum drying to eliminate solvent traces, resulting in a dry residue. This dry residue was subsequently dissolved in 100 μL of high-purity methanol, ensuring complete dissolution of the compound. HPLC analysis was performed using an Agilent 1200 Infinity instrument equipped with a diode-array detector, which operated at a wavelength of 310 nm for compound 7D detection. Chromatographic separation was achieved using a C4 column (Teknokroma, Tracer Excel C4 120, 25 mm × 0.46 cm, 5 μm particle size). A linear gradient elution method was employed, with the mobile phase consisting of acetonitrile and water containing 0.1% formic acid. The gradient ranged from 2% to 100% acetonitrile over an 18-min period. The flow rate was maintained at 1 mL/min, and the column temperature was set at 25 °C. Each sample, consisting of 100 μL, was injected into the HPLC system. Data analysis involved the integration of chromatographic peaks corresponding to compound 7D. The area under each peak was determined, and the concentration of the compound was quantified using the constructed standard curve, generated from the known concentrations of the working solutions.

## Ethics

**Small animals**: All the mice strains and experimental protocols were approved by the institutional animal ethics committee of regional center for Biotechnology (reference number RCB/IAEC/2020/065). The housing and husbandry conditions and gender of animals involved in experiments is reported according to the protocols of the Committee for Control and Supervision of Experiments on Animals (CCSEA), a statutory Committee of Department of Animal Husbandry and Dairying (DAHD), Govt. of India. All the experiments were conducted following the approved protocols in small animal facility (SAF), NCR biotech science cluster.

## The paper explained

### Problem

Effective vaccines against dengue virus (DENV) are limited and there has been significant focus on the development of effective anti-viral against the disease. We recently reported that platelet factor 4 (PF4 or CXCL4), primarily released from activated platelets, promotes DENV infection in patients. CXCL4 inhibits interferon (IFN)$\alpha/\beta$ synthesis by inhibiting CXCR3:p38 pathway in vitro.

### Results

In a concurrent in silico search for other CXCR3-antagonists, we identified 7D as a promising candidate from our in-house library, capable of inhibiting all four serotypes of DENV. 7D supplementation (8 mg/kg body weight) to DENV2-infected mice improved synthesis of IFN-$\alpha/\beta$ and IFN-$\lambda$ via CXCL4:CXCR3:p38:IRF3 pathway and rescued disease symptoms like thrombocytopenia and leukopenia, decreased vascular-leakage and increased survival. Besides, having the inhibiting property to deacetylase Sirt-1, 7D promoted acetylation and phosphorylation of STAT3, in turn increased proliferating plasmablasts and germinal centre maturation, and generation of neutralizing antibodies against DENV2 in mice. A half-life of ~2.85 h in mice plasma and no significant toxicity suggest the safe usage of 7D in vivo.

### Impact

Together, our studies identify compound 7D as a stimulator of IFN$\alpha/\beta/\lambda$ synthesis via CXCL4:CXCR3:p38:IRF3 pathway and also a booster for neutralizing-antibodies generation by promoting STAT3 acetylation in plasmablasts, capable of protecting dengue infection of all serotypes.

**Human subjects**: Blood samples from healthy participants were collected under the approved protocol from the Institutional Ethics Committee (reference number RCB-BBB-IEC-H-35). PBMCs were isolated using the Ficoll-Paque density gradient centrifugation for further experiments. Recruitment of human participants follows the principles set out in the WMA Declaration of Helsinki and the Department of Health and Human Services Belmont Report.

## Statistical analysis

Data are presented as mean ± SEM (standard error mean) or median ± IQR (Inter quartile range) of at least three independent experiments. Statistical comparisons between experimental groups with parametric data were analyzed either using an unpaired *t* test or one-way ANOVA followed by Bonferroni's for correction for multiple comparisons. Non-parametric data was analyzed by Mann–Whitney *U* test followed by Kruskal–Wallis test in case of multiple comparison. D'Agostino–Pearson test was used to analyze the distribution of data. GraphPad Prism version 8.0 software was used to analyze the data and *P* values < 0.05 were considered statistically significant.

## Data availability

The gene sequence data presented in this study can be found in online repositories, NCBI, Gene Bank PQ198062.

The source data of this paper are collected in the following database record: biostudies:S-SCDT-10_1038-S44321-024-00137-8.

## Peer review information

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

## Acknowledgements

Authors acknowledge the funding by grants: BT/PR22881 and BT/PR22985 from the Department of Biotechnology, Govt. of India to PG; CRG/000092 from the Science and Engineering Research Board, Govt. of India to PG and SA; and Department of Science and Technology, Govt. of India for DST-INSPIRE Fellowship (2014/113) to NP. Authors acknowledge the DENV virus strains as the kind gifts from Dr. Ashley L. St. John, Duke-NUS Medical School, Singapore; and Sirt1$^{-/-}$ MEFs as the gifts from Dr. Michale Mcburney, Ottawa Hospital Research Institute, Canada, and Dr. Ullas Kolthur-Seetharam, Centre for DNA Fingerprinting & Diagnostics, Hyderabad, India. Authors acknowledge the help of Dr. Arundhati Tiwari of the Regional Centre for Biotechnology (RCB), Faridabad, India, for editing. Authors thank to Dr. Anna George of the RCB, Faridabad; and Dr. Anmol Chandele of ICGEB, New Delhi, India for scientific advice.

## Author contributions

**Kishan Kumar Gaur**: Data curation; Software; Formal analysis; Investigation; Visualization; Methodology; Writing—review and editing. **Tejeswara Rao**

**Asuru**: Data curation; Software; Formal analysis; Investigation; Visualization; Methodology; Writing—review and editing. **Mitul Srivastava**: Software; Formal analysis; Methodology. **Nitu Singh**: Validation; Investigation; Methodology. **Nikil Purushotham**: Validation; Investigation; Methodology. **Boja Poojary**: Resources; Validation; Methodology. **Bhabatosh Das**: Software; Validation; Investigation; Writing—review and editing. **Sankar Bhattacharyya**: Conceptualization; Funding acquisition; Methodology; Writing—review and editing. **Shailendra Asthana**: Conceptualization; Resources; Data curation; Supervision; Funding acquisition; Investigation; Project administration; Writing—review and editing. **Prasenjit Guchhait**: Conceptualization; Resources; Data curation; Formal analysis; Supervision; Funding acquisition; Investigation; Methodology; Writing—original draft; Project administration.

Source data underlying figure panels in this paper may have individual authorship assigned. Where available, figure panel/source data authorship is listed in the following database record: biostudies:S-SCDT-10_1038-S44321-024-00137-8.

## Disclosure and competing interests statement

The authors declare no competing interests.

# Expanded View Figures

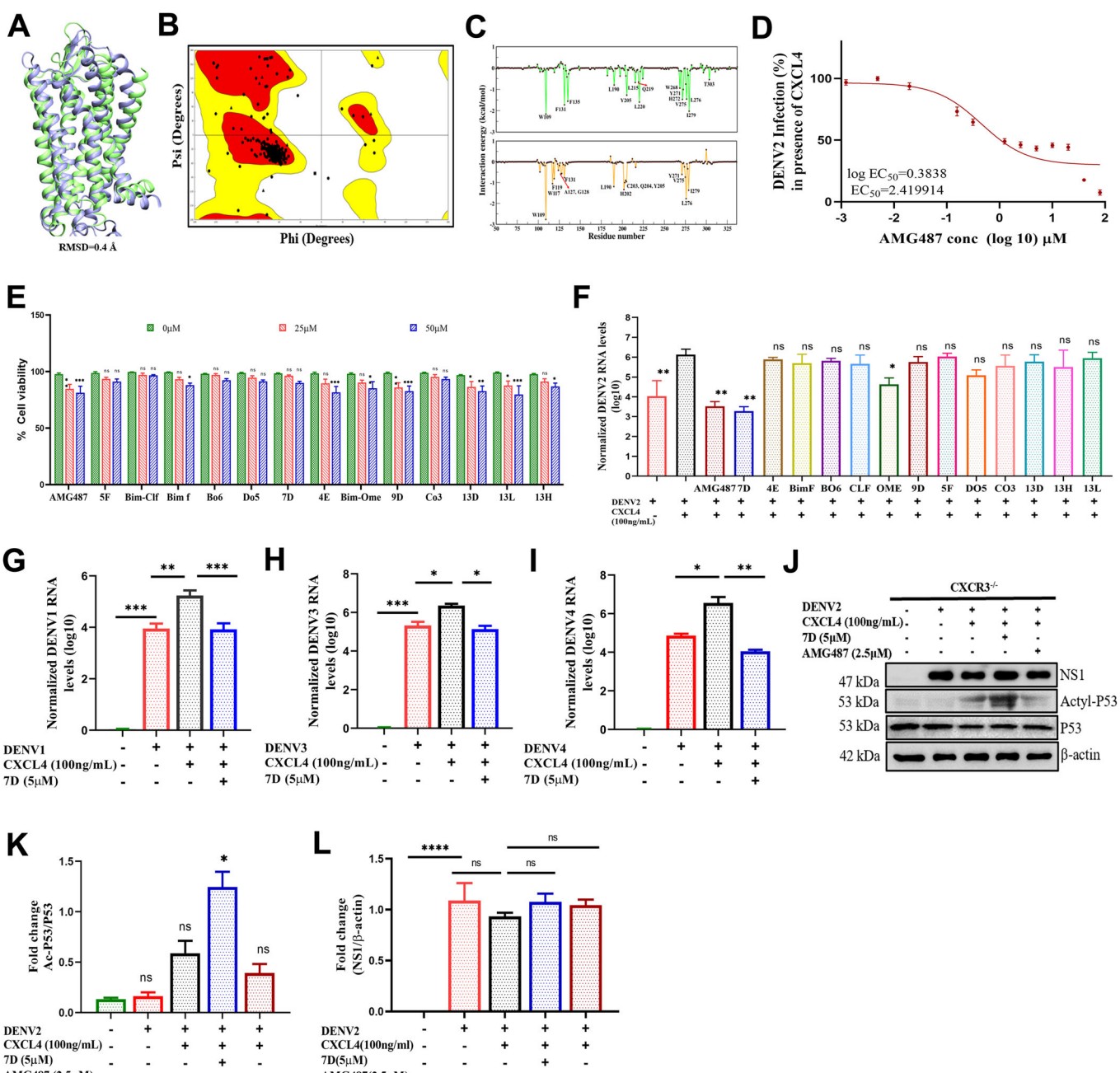

**Figure EV1. In silico drug design and viral replication data related to main Fig. 1.**

(A) Superimposed structures of crystal CCR5 (blue) and model of CXCR3 (green). (B) Ramachandran plot of CXCR3 homology model (C) CXCR3-7D (green) and CXCR3-AMG487 (yellow) binding site interaction energies of residues. Related data of Fig. 1J,K. (D) U937-DC-SIGN cells were infected with MOI ~1 of DENV2 (P23085 INDI-60) in presence CXCL4 (100 ng/ml) and various concentrations of AMG487. The $EC_{50}$ value of AMG487 was calculated from the dose-response curve from drug effects on viral genome (measured by qRT-PCR) from independent experiments, $n = 3$. (E) Cell viability was measured after treatment with all 13 compounds from in silico hits. U937-DC-SIGN cells were incubated with the compounds for 48 h and cell viability were measured using MTT assay, $n = 3$ independent experiments, two-way ANOVA was used ($P$ values: 0.004; 0.0004; 0.01; 0.0002; 0.005; 0.005; 0.0004; 0.02; 0.0019; 0.01; 0.0001; 0.01). (F) Effects of 13 compounds (2.5 μM) on DENV2 replication in U937-DC-SIGN cells was measured in presence of CXCL4 (100 ng/ml). Viral genome was quantified using qRT-PCR from independent experiments, n = 3, one-way ANOVA and Kruskal–Wallis test was used ($P$ values: 0.008; 0.003; 0.0014; 0.02). (G–I) U937-DC-SIGN cells were infected with MOI ~1 of (G) DENV1 (H) DENV3 and (I) DENV4 (strain H241) in presence of CXCL4 (100 ng/ml) and 7D (5 μM) and viral genome was quantified using qRT-PCR from independent experiments, $n = 3$, one-way ANOVA and Bonferroni's post-test were used ($P$ values: G: 0.001; 0.0014; 0.0008, H: 0.001; 0.01; 0.02, I: 0.01; 0.003). (J–L) Western blot was performed for (J) NS1, actyl-p53, p53 and β-actin in lysate from CXCR3$^{-/-}$ monocytes from Fig. 2G. Densitometry of (K) actyl-p53 and (L) NS1 blots, $n = 3$ independent experiments, one-way ANOVA and Bonferroni's post-test were used ($P$ value: K: 0.01, L: 0.0001). Data information: (D–I, K, L) Data are mean ± SEM, *$P < 0.05$, **$P < 0.01$, ***$P < 0.001$, ****$P < 0.0001$, ns=non-significant.

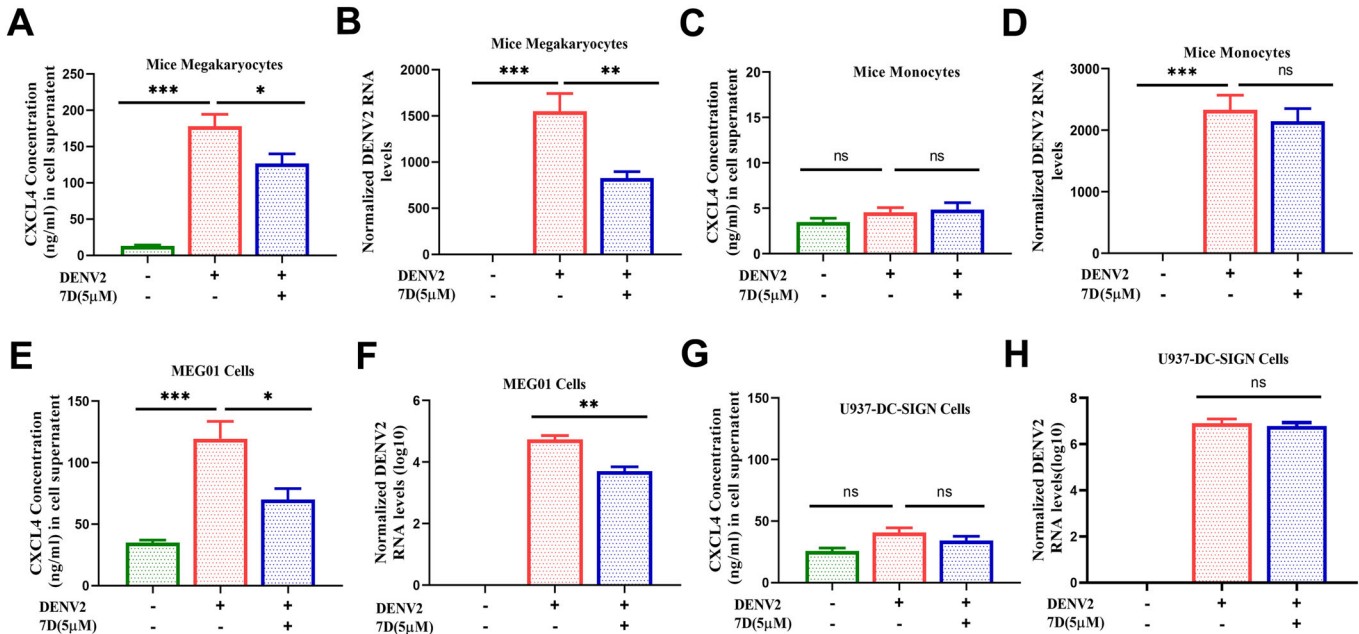

**Figure EV2. Effects of 7D on DENV2 replication on megakaryocytes in vitro.**

(A, B) Megakaryocytes isolated from mice bone marrow, and infected with DENV2 (MOI ~ 1) for 24 h with and without 7D. Culture supernatant was used for measuring (A) CXCL4 levels using ELISA, and (B) cell pallet was used for detecting viral genome using qRT PCR, $n = 3$ independent experiments for both (P values: **A**: 0.0002; 0.04, **B**: 0.0002; 0.008). (C, D) Similar experiment was performed in monocytes (as CXCL4 non-producing cells) isolated from whole blood of these mice. (C) CXCL4 and (D) DENV2 genome were detected, $n = 3$ independent experiments for both assays (P value: 0.0002). A similar experiment was performed in human MEG-01 cell line (CXCL4 producing cells), $n = 3$ independent experiments (P values: **E**: 0.0008; 0.01, **F**: 0.006) and (G, H) U937-DC-SIGN cell line (CXCL4 non-producing cells), $n = 3$ independent experiments for above assays. One-way ANOVA and Bonferroni's post-test were used for data analysis. Data information: (**A–E**, **G**) One-way ANOVA and Bonferroni's post-test were used for data analysis. Data are mean ± SEM. (**F**, **H**) Student's t-test was used. Data are mean ± SEM. *$P < 0.05$, **$P < 0.01$, ***$P < 0.001$, ns=non-significant.

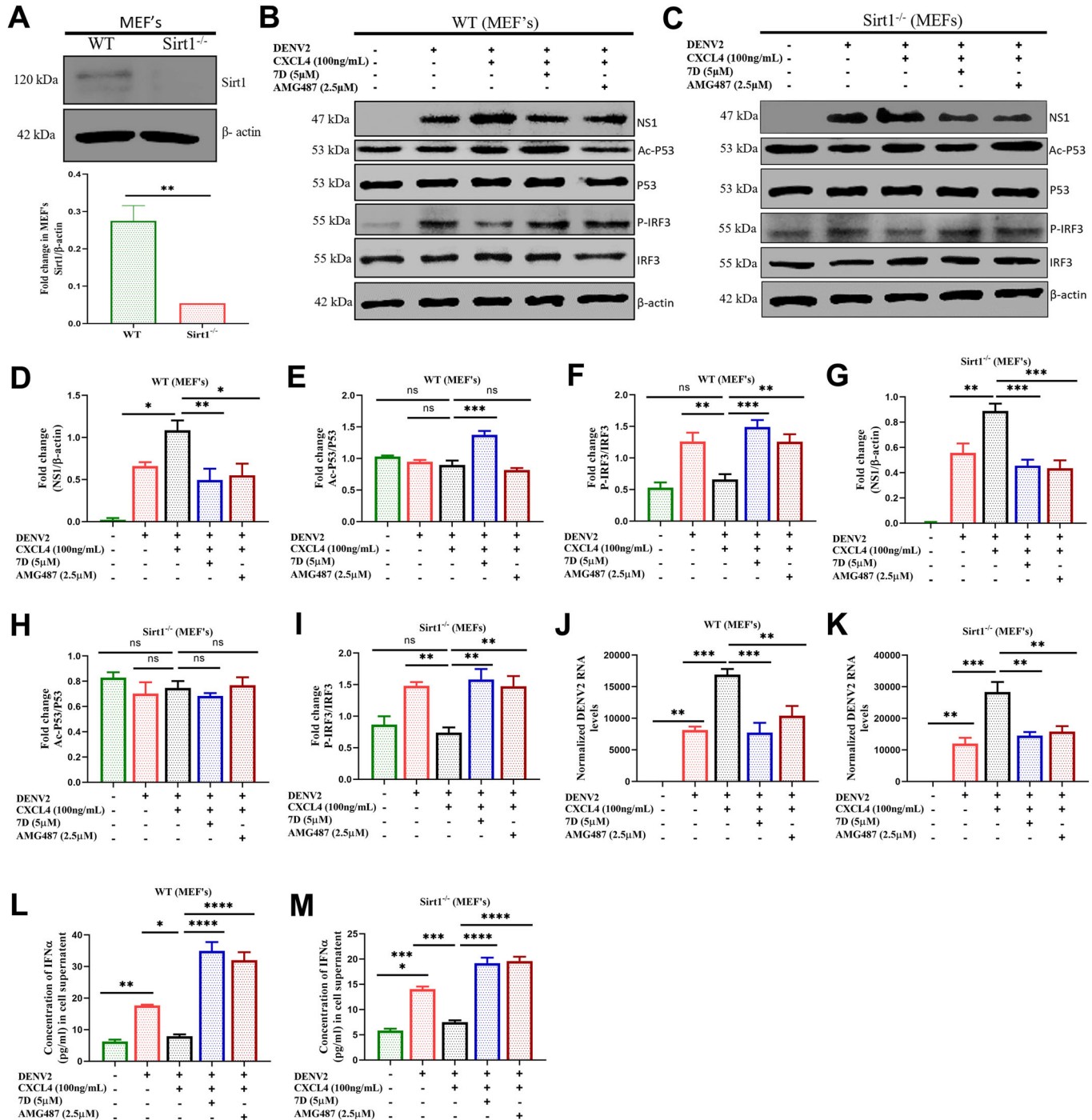

**Figure EV3. Effects of 7D on DENV2 replication in Sirt1$^{-/-}$ MEFs in vitro.**

(A) Sirtuin-1 (Sirt1) expression in Sirt1$^{-/-}$ and WT mouse embryonic fibroblasts (MEFs) in western blot assay. Densitometry analysis independent experiments $n = 3$, Student's $t$ test was used ($P$ value: 0.005). Above cells were infected with DENV2 and treated with 7D as mentioned for U937-DC-SIGN in Fig. 1F. After experiment, (B) WT and (C) Sirt1$^{-/-}$ cells were used for western blot analysis for NS1, Ac-P53 and P-IRF3. (D–I) Densitometry data from, $n = 3$ independent experiments, one-way ANOVA and Bonferroni's post-test were used ($P$ values: D: 0.04; 0.007; 0.01, E: 0.001, F: 0.009; 0.0002; 0.009, G: 0.005; 0.0008; 0.0006, I: 0.006; 0.002; 0.007). (J, K) Viral genome was quantified in WT and Sirt1$^{-/-}$ MEFs pellet using qRT-PCR, $n = 3$ independent experiments, one-way ANOVA and Bonferroni's post-test were used ($P$ values: J: 0.0013; 0.0007; 0.0005; 0.006, K: 0.004; 0.0004; 0.002; 0.003). (L, M) IFNα level was measured using ELISA from cell supernatant of above experiments, $n = 3$ independent experiments, one-way ANOVA and Bonferroni's post-test were used ($P$ values: L: 0.003; 0.01; 0.0001; 0.0001, M: 0.0001; 0.0002; 0.0001; 0.0001). Data information: (A, D–M) Data are mean ± SEM, *$P < 0.05$, **$P < 0.01$, ***$P < 0.001$, ****$P < 0.0001$, ns=non-significant.

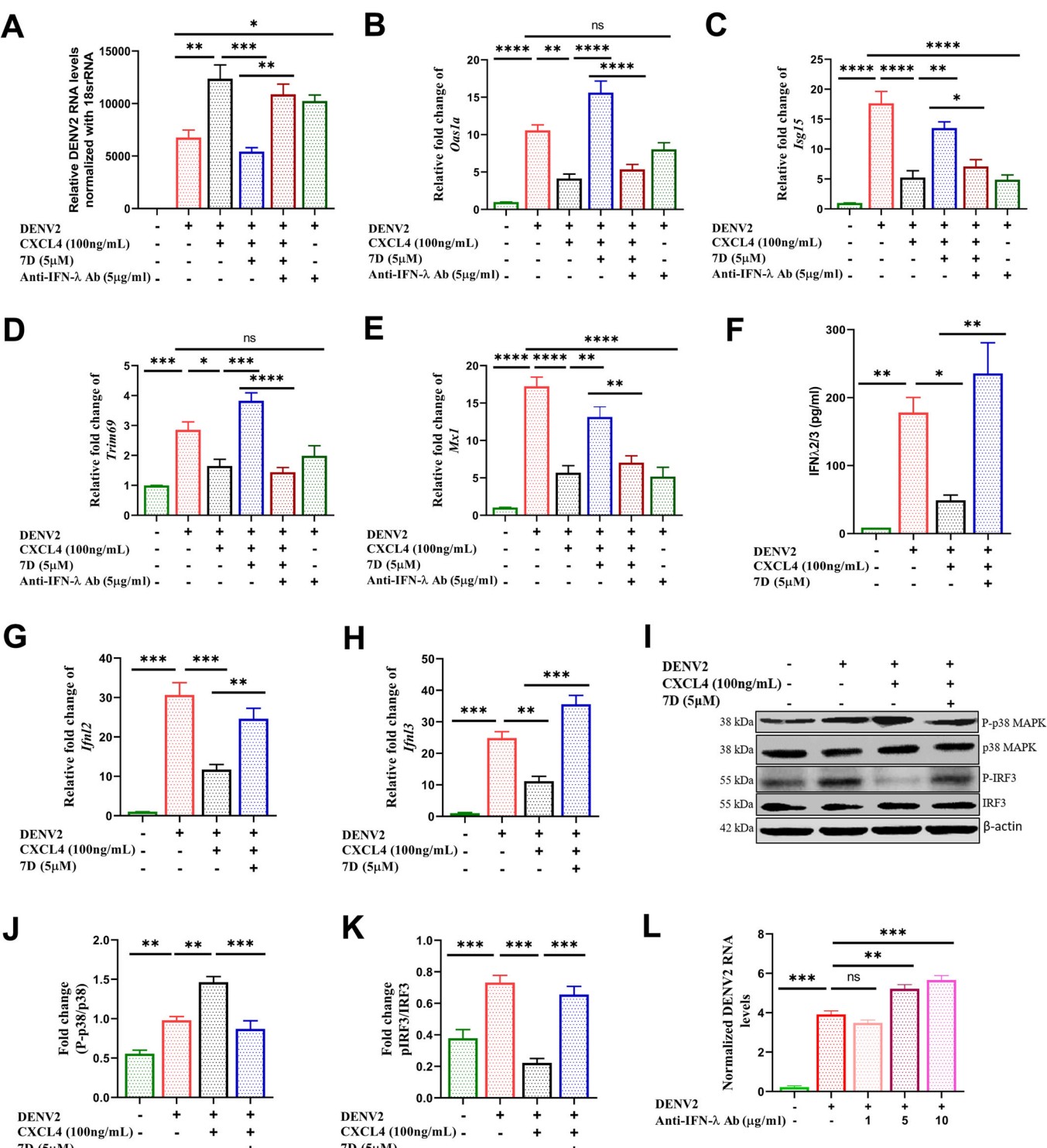

**Figure EV4.  Effects of 7D on DENV2 replication in BMDMs of AG129 mice.**

DENV2 infection experiment was performed in bone marrow derived macrophages (BMDMs), isolated from AG129 mice, in presence of 7D. A blocking antibody to IFNλ2/3 (5 μg/ml, standardization of working concentration is mentioned in Fig. EV4L below) was used to investigate the effects of type-III IFN. (A) Viral genome was quantified in cell pellets using qRT-PCR, $n = 3$ independent experiments (P values: 0.0011; 0.0002; 0.03; 0.0014). (B–E) Relative gene expressions of interferon-stimulated genes (ISGs), (B) Oas1a, (C) Isg15, (D) Trim69, and (E) Mx1 were measured from above experiments, $n = 3$ independent experiments, (P values: **B**: 0.0001; 0.0013; 0.0001; 0.0001, **C**: 0.0001; 0.0001; 0.0012; 0.01; 0.0001, **D**: 0.0006; 0.01; 0.0008; 0.0001, **E**: 0.0001; 0.0001; 0.0014; 0.009; 0.0001). (F) IFN λ2/3 levels were measured in the supernatant of these cells using ELISA, $n = 3$ independent experiments, (P values: 0.004; 0.02; 0.002). (G, H) Relative gene expressions of *ifnl2* and *ifnl3* were quantified from the above cell pellets, $n = 3$ independent experiments, (P values: **G**: 0.001; 0.001; 0.008, **H**: 0.001; 0.003; 0.001). (I–K) Western blot analysis for P-p38:p38 and P-IRF3:IRF3. (J, K) Densitometry data from, $n = 3$ independent experiments, (P values: **J**: 0.007; 0.003; 0.001, **K**: 0.0006; 0.0002; 0.0003). (L) Concentration-dependent effect of blocking antibody against type-III IFN was tested on viral replication. DENV2 mRNA was quantified from pellets of DENV2-infected U937-DC-SIGN cells in presence of increasing concentration (1, 5 and 10 μg/ml) of blocking antibody against IFN-λ2/3 Ab, $n = 3$ independent experiments, (P values: 0.001; 0.002; 0.0002). One-way ANOVA and Bonferroni's post-test were used for above analysis. Data information: (**A–H**, **J–L**) One-way ANOVA and Bonferroni's post-test were used. Data are mean ± SEM, *$P < 0.05$, **$P < 0.01$, ***$P < 0.001$, ****$P < 0.0001$ and ns=non-significant.

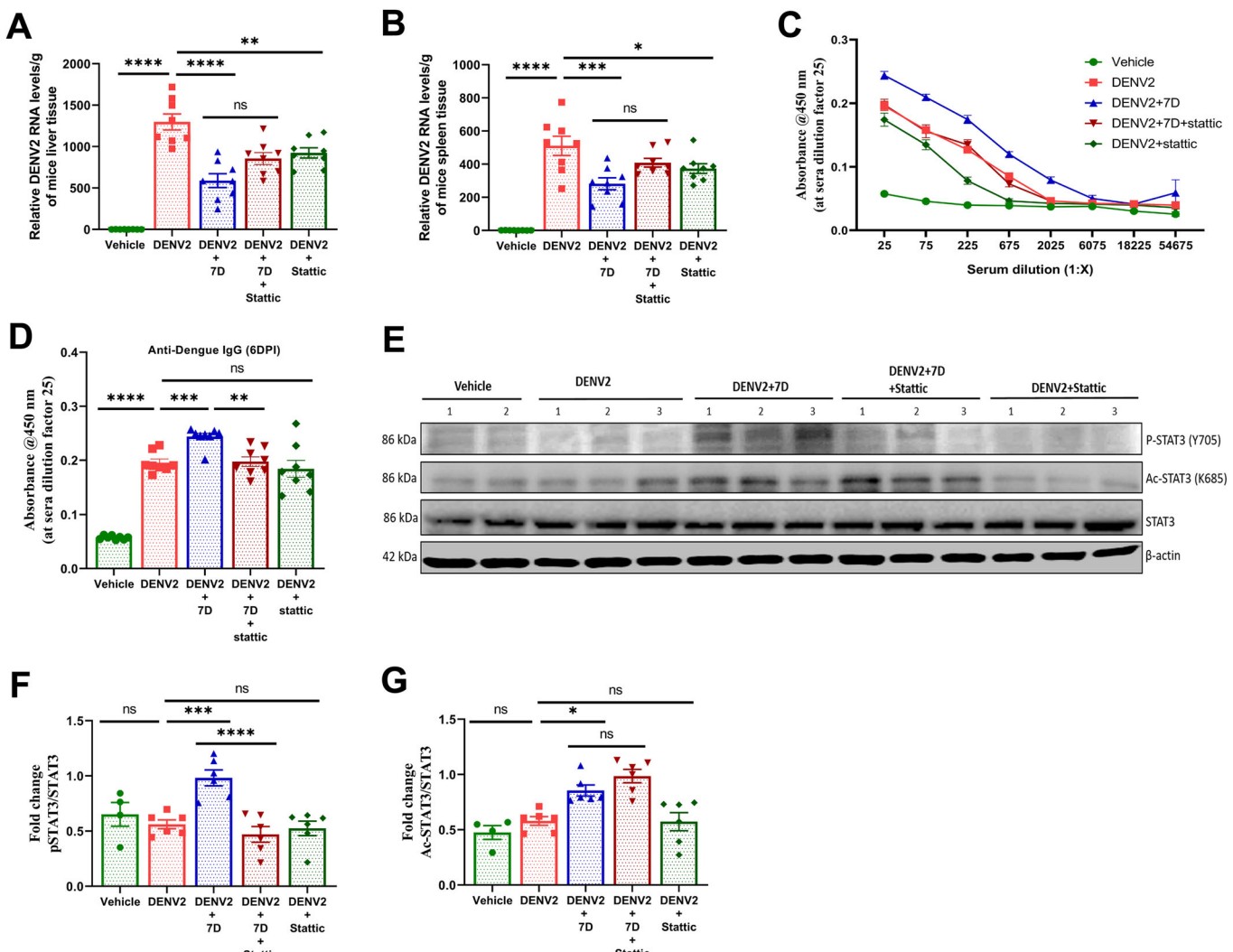

**Figure EV5.  Effect of STAT3-inhibitor on 7D-mediated antibody generation in DENV2-infected AG129 mice.**

As mentioned in the Fig. 4A, a similar experiment was performed in AG129 mice treated with 7D and STAT3-inhibitor Stattic (10 mg/kg/body weight, referred concentration from vendor's manual). (A, B) DENV2 viral RNA was quantified in liver and spleen respectively using qRT-PCR, $n = 8$ mice per group, (P values: A: 0.0001; 0.0001; 0.003, B: 0.0001; 0.0003; 0.04). (C, D) Anti-dengue IgG was measured from the serum of mice using ELISA, $n = 8$, (P values: 0.0001; 0.0005; 0.003). (E–G) Ac-STAT3 and P-STAT3 were measured in spleenocytes by western blot and normalized with total STAT3. Densitometry of the above blots, $n = 4$ vehicle, $n = 6$ mice per group (P values: F: 0.0008; 0.0001, G: 0.01). One-way ANOVA and Bonferroni's post-test were used for all above analysis. Data information: (A–D, F, G) One-way ANOVA and Bonferroni's post-test were used. Data are mean ± SEM, *$P < 0.05$, **$P < 0.01$, ***$P < 0.001$, ****$P < 0.0001$, ns=non-significant.

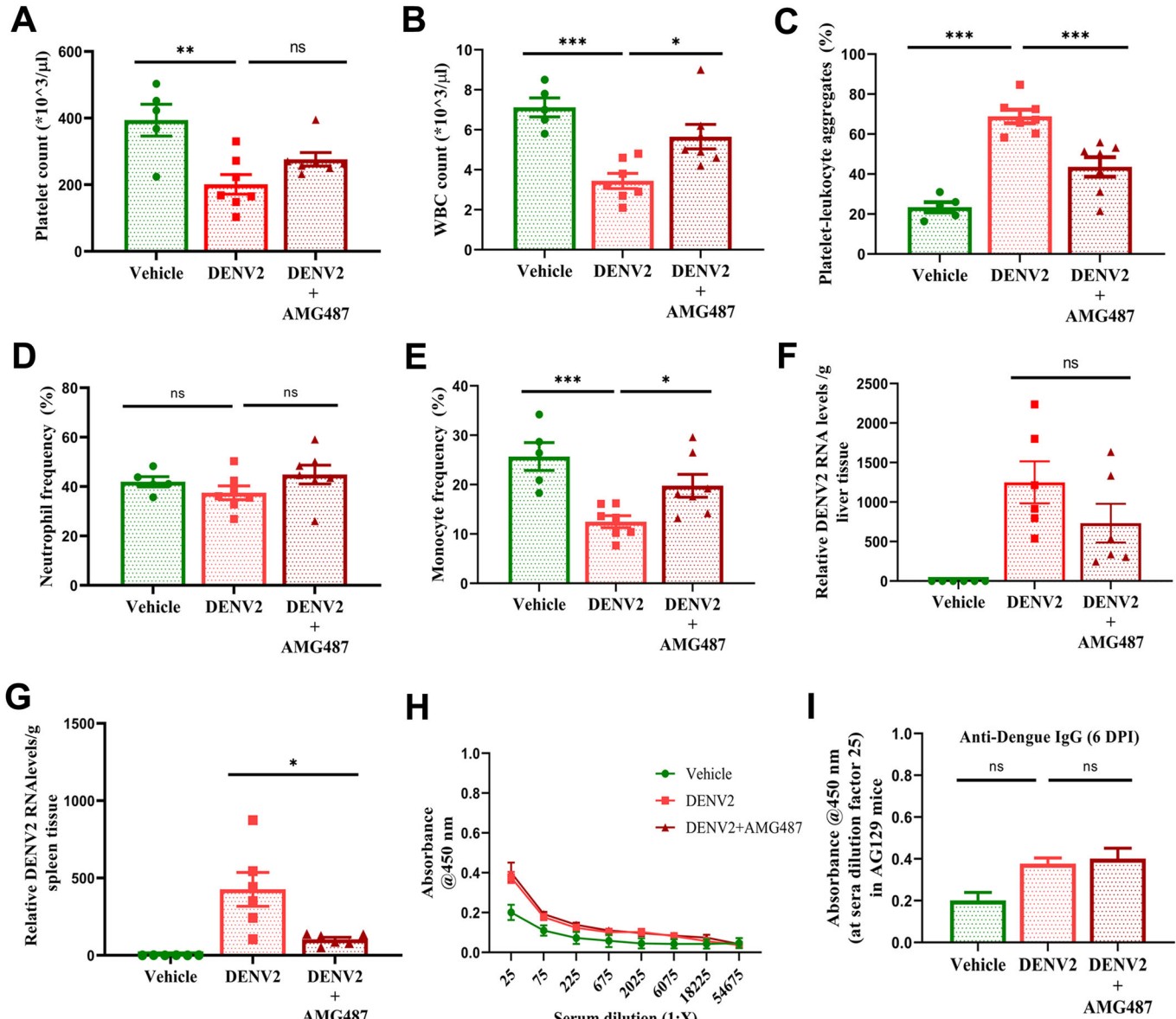

**Figure EV6.  Effects of AMG487 on DENV2 infection in AG129 mice.**

AG129 mice were infected with DENV2 and treated with AMG487 (8 mg/kg body weight, referred concentration from our previous work). (**A**) Platelets and (**B**) WBCs were counted in whole blood using hematology analyzer. (**C**) Platelet-leukocyte aggregates, (**D**) neutrophils and (**E**) monocytes percentage were measured using flow cytometry, $n = 5$ vehicle, $n = 7$ mice per group, one-way ANOVA and Bonferroni's post-test were used for above analysis, ($P$ values: **A**: 0.0015; **B**: 0.0003; 0.01; **C**: 0.001; 0.0005; **E**: 0.001; 0.03). (**F, G**) DENV2 genome was quantified by qRT-PCR in (**F**) liver and (**G**) spleen, $n = 6$ mice per group and Mann–Whitney $U$ test was used, ($P$ value: 0.01). (**H, I**) IgG against DENV2 antigen was measured in mice plasma, ($n = 6$ per group and one-way ANOVA and Kruskal–Wallis test was used). Sera dilution factor 25 at OD$_{450nm}$. Data information: (**A–E, I**) Data are mean ± SEM, and (**F, G**) data are median ± IQR, $^*P < 0.05$, $^{**}P < 0.01$, $^{***}P < 0.001$, ns=non-significant.

