## [Peer Review File · EMBO Molecular Medicine]

7D, a small molecule inhibits dengue infection by increasing interferons and neutralizing antibodies

Kishan Gaur, Tejeswara Asuru, Mitul Srivastava, Nitu Singh, Nikil Purushotham, Boja Poojary, Bhabatosh Das, Sankar Bhattacharyya, Shailendra Asthana, and Prasenjit Guchhait

Corresponding authors: Prasenjit Guchhait (prasenjit@rcb.res.in) , Shailendra Asthana (sasthana@thsti.res.in)

Review Timeline:

Submission Date:	30th Dec 23
Editorial Decision:	9th Feb 24
Revision Received:	30th Apr 24
Editorial Decision:	10th Jun 24
Revision Received:	3rd Aug 24
Editorial Decision:	15th Aug 24
Revision Received:	27th Aug 24
Editorial Decision:	27th Aug 24
Revision Received:	28th Aug 24
Accepted:	29th Aug 24

Editor: Poonam Bheda

Transaction Report:

9th Feb 2024

Dear Dr. Guchhait,

Thank you for the submission of your manuscript to EMBO Molecular Medicine. We have now received feedback from the three reviewers who agreed to evaluate your manuscript. As you will see from the reports below, the referees acknowledge the interest of the study; however they also have concerns on the mechanistic interpretations and the lack of evidence that 7D exerts its effect directly through CXCR3, for which additional experimental validation is needed to fully support the conclusions.

In particular, they suggest further investigation as to whether 7D actually inhibits DENV2 via IFN signaling (particularly in AG129 that lack IFN receptors), and also that you provide evidence that 7D functions as a CXCR3 inhibitor by testing the effect of 7D on DENV2 replication in CXCR3 knock-outs.

Addressing the reviewers' concerns in full in a point-by-point response will be necessary for further considering the manuscript in our journal, and acceptance of the manuscript will entail a second round of review. EMBO Molecular Medicine encourages a single round of revision only and therefore, acceptance or rejection of the manuscript will depend on the completeness of your responses included in the next, final version of the manuscript. For this reason, and to save you from any frustrations in the end, I would strongly advise against returning an incomplete revision. If you would like to discuss further the points raised by the referees, I am available to do so via email or video. Let me know if you are interested in this option.

We are expecting your revised manuscript within three months, if you anticipate any delay, please contact us. When submitting your revised manuscript, please carefully review the instructions that follow below. We perform an initial quality control of all revised manuscripts before re-review; failure to include requested items will delay the evaluation of your revision.

We require:

4) A .docx formatted letter INCLUDING the reviewers' reports and your detailed point-by-point responses to their comments. As part of the EMBO Press transparent editorial process, the point-by-point response is part of the Review Process File (RPF), which will be published alongside your paper.

5) A complete author checklist, which you can download from our author guidelines (<https://www.embopress.org/page/journal/17574684/authorguide#submissionofrevisions>). Please insert information in the checklist that is also reflected in the manuscript. The completed author checklist will also be part of the RPF.

6) Please note that all corresponding authors are required to supply an ORCID ID for their name upon submission of a revised manuscript.

7) It is mandatory to include a 'Data Availability' section after the Materials and Methods. Before submitting your revision, primary datasets produced in this study need to be deposited in an appropriate public database, and the accession numbers and database listed under 'Data Availability'. Please remember to provide a reviewer password if the datasets are not yet public (see <https://www.embopress.org/page/journal/17574684/authorguide#dataavailability>).

In case you have no data that requires deposition in a public database, please state so in this section. Note that the Data Availability Section is restricted to new primary data that are part of this study. This study includes no data deposited in external repositories.

8) For data quantification: please specify the name of the statistical test used to generate error bars and P values, the number (n) of independent experiments (specify technical or biological replicates) underlying each data point and the test used to calculate p-values in each figure legend. The figure legends should contain a basic description of n, P and the test applied. Graphs must include a description of the bars and the error bars (s.d., s.e.m.). Please provide exact p values.

9) Our journal encourages inclusion of *data citations in the reference list* to directly cite datasets that were re-used and

obtained from public databases. Data citations in the article text are distinct from normal bibliographical citations and should directly link to the database records from which the data can be accessed. In the main text, data citations are formatted as follows: "Data ref: Smith et al, 2001" or "Data ref: NCBI Sequence Read Archive PRJNA342805, 2017". In the Reference list, data citations must be labeled with "[DATASET]". A data reference must provide the database name, accession number/identifiers and a resolvable link to the landing page from which the data can be accessed at the end of the reference. Further instructions are available at .

10) We replaced Supplementary Information with Expanded View (EV) Figures and Tables that are collapsible/expandable online. A maximum of 5 EV Figures can be typeset. EV Figures should be cited as "Figure EV1, Figure EV2" etc... in the text and their respective legends should be included in the main text after the legends of regular figures.

13) Author contributions: CRediT has replaced the traditional author contributions section because it offers a systematic machine readable author contributions format that allows for more effective research assessment. Please remove the Authors Contributions from the manuscript and use the free text boxes beneath each contributing author's name in our system to add specific details on the author's contribution. More information is available in our guide to authors.

Please also suggest a striking image or visual abstract to illustrate your article as a PNG file 550 px wide x 300-600 px high. Share synopsis text and image, as well as eTOC:

Please note that these would be the final versions and changes during proofing are usually not allowed

16) As part of the EMBO Publications transparent editorial process initiative (see our policy here: https://www.embopress.org/transparent-process#Review_Process), EMBO Molecular Medicine will publish online a Peer Review File (PRF) to accompany accepted manuscripts.

In the event of acceptance, this file will be published in conjunction with your paper and will include the anonymous referee reports, your point-by-point response and all pertinent correspondence relating to the manuscript. Let us know whether you agree with the publication of the PRF and as here, if you want to remove or not any figures from it prior to publication.

I look forward to receiving your revised manuscript.

Yours sincerely,

Poonam Bheda

Poonam Bheda, PhD
Scientific Editor
EMBO Molecular Medicine

**** Reviewer's comments ****

Referee #1 (Comments on Novelty/Model System for Author):

For explain the immune mechanisms, the AG129 mice is inappropriate.

Referee #1 (Remarks for Author):

The manuscript by Kishan Gaur et al. described a CXCR3-antagonist, named 7D, that inhibits CXCL4-mediated dengue infection in vitro. Also, 7D showed an inhibitory effect on DENV infection in two mouse models. Although the antiviral effects of 7D are obvious, the potential mechanisms are very confusing. There are several major flaws in the mechanism part of this manuscript.

Major concerns:

In vitro study:

1. For the in vitro study. 7D only showed inhibitory effect in the presence of CXCL4. How about the impact of 7D on DENV infection without adding of CXCL4? Especially in immune-derived cells, which supposed to express CXCL4 after DENV infection.
2. 7D may increase IFN expression by the CXCL4: CXCR3: p38: IRF3 pathway. Does 7D also suppress DENV infection in IFN-deficient cells? If that is true, what's the mechanism?
3. All the in vitro study data are shown as bars, but no spots indicate the number of samples or the number of biological repeats.

In vivo study:

1. If the antiviral effect of 7D is CXCR3 dependent, the authors should include the experiments using CXCR3^{-/-} mice or using CXCR3 blocking reagent to confirm the results.
2. AG129 mice lack IFN (I/II) receptors; the elevated IFN levels will have no effect on viral replication in these mice. Thus, the elevated IFNs will not be the mechanism for 7D inhibits DENV in AG129 mice.
3. I agree that the elevated Ab titers in 7D- treated mice could be a reason for inhibiting DENV, but why? The authors try to show that 7D increased the Acetylation and phosphorylation of STAT3. But what are the mechanisms behind this? Is it CXCR3 dependent? Also, in CXCR3^{-/-} mice, does 7D increase DENV-induced antibody production?
4. The authors cited a reference to show 7D may increase the Acetylation of STAT3 by inhibiting Sirt1, but did not show any data on this during DENV infection. Does 7D inhibit DENV infection in Sirt1 or STAT3-deficient mice?

Generally, the two mechanisms for 7D inhibit DENV in vivo are very unclear. There is a lot of work to do before the manuscript can be accepted for publication.

Referee #2 (Comments on Novelty/Model System for Author):

See comments to author.

Referee #2 (Remarks for Author):

This is a comprehensive study that identifies a novel antagonist of CXCR3, through in silico screening, followed by testing in various in vitro and in vivo systems to demonstrate the small molecule is effective in modulating signaling to dengue infection and that it can enhance interferon production in vivo, while reducing infection. The authors also characterize the effects of the drug on cellular immune responses and antibodies following infection. The use of CXCR3-KO cells to demonstrate the mechanism of drug is through this receptor is an elegant supporting study. The use of both IFN deficient and WT animals is also an advantage of the study since the results were consistent. The read-outs in vivo are quite broad, and more will need to be

done (in the future, not necessarily for this first manuscript) to define the mechanisms through which the drug works in vivo to alter T cell and antibody responses. Although some signaling is presented I am not sure this is the whole mechanism, and the potential for off-target effects were not thoroughly investigated, so perhaps some discussion of those points would improve the manuscript.

Specific suggestions for improvements:

Perhaps a draw-back of the study is that it is not necessarily shown that reduced TNF expression is beneficial in dengue. Although TNF has been associated with vascular leak in some models, the use of T cell derived TNF has not been explicitly investigated. It should also be noted that TNF has beneficial roles in adaptive immune responses aside from inducing vascular permeability.

The authors should check that the word "significantly" is used to mean statistical significance only rather than to highlight compelling conclusions.

I am not sure if the title is clear what 7D is- Maybe the words "small molecule" could be added to the title?

This statement needs a citation: "The increased CD4CD8 double positive T cells is considered to be associated with the severity of dengue." It is a bit unexpected that the authors chose to focus on double positive cells. Panels 5A-B should be described in the text as well. It's not specifically stated that those populations did not differ.

For the figures, all of the labels - e.g. A, B, C are quite small. Figure 3 would also benefit from aligning the panels more accurately.

The figure 4 legend should indicate which panels were performed in WT mice, since this figure apparently contains both data from AG129 and WT mice.

Referee #3 (Comments on Novelty/Model System for Author):

AG129 and WT mouse models are fine

Referee #3 (Remarks for Author):

In this manuscript titled "7D inhibits dengue infection by increasing synthesis of IFN α/β and neutralizing antibodies via CXCL4:CXCR3:p38:IRF3 and Sirt1:STAT3 axes respectively", Kishan Kumar Gaur et al. identify a CXCR3 inhibitor by in silico modeling, i.e., 7D. They show that 7D inhibits CXCL4-enhanced DENV2 replication in U937-DC-SIGN cell and primary mouse monocytes. Using IFN $\alpha/\beta/\gamma$ R $^{-/-}$ AG129 and WT mice, the authors show that 7D is safe and potent to suppress DENV2 replication, it renders animals resistant to lethal DENV2 infection in AG129 mice, reduces vascular leakage, thrombocytopenia, and leukopenia. Mechanistically, 7D promotes type I IFN expression by suppressing the CXCL4:CXCR3:p38:IRF3 pathway, enhances STAT3-acetylation in plasma cells thus neutralizing antibodies. Another potent CXCR3 inhibitor, AMG 487, promotes type I IFN expression, protects AG119 mice from DENV2 infection, but has no effect on B cells and neutralizing antibodies.

Overall, the study identifies a new CXCR3 inhibitor that might be therapeutic against DENV2 infection. However, the mechanism of action of 7D requires more rigorous and direct evidence.

Major critiques

1. The fact that AMG487 still inhibits DENV2 replication in AG129 mice without increasing neutralizing antibodies suggests that the primary antiviral mechanism of AMG487 is independent of the type I IFN response. Different from AMG487, 7D promotes both type I IFN and neutralizing antibody production, the latter may inhibit DENV replication in AG129 mice in the absence of IFN-I signaling. The authors need to strengthen that 7D and AMG487 indeed inhibits DENV2 via type I IFNs using AG129 monocytes ex vivo.
2. Small molecule inhibitors may have many off-target effects in vivo. The authors should provide direct evidence that CXCR3 signaling facilitates DENV2 and that 7D inhibits DENV2 infection via CXCR3 in vivo, since the Cxcr3 $^{-/-}$ animals are available (Fig.2). Are Cxcr3 $^{-/-}$ animals resistant to DENV2 infection and is the effect of 7D lost in Cxcr3 $^{-/-}$ mice? Do Cxcr3 $^{-/-}$ mice have more neutralizing antibodies?
3. Fig.6. It is not clear about the time of sampling, presumably Day 6 p.i. based on Fig.4A. This reviewer is not sure DENV2-specific IgG is significantly induced and is the primary antiviral mechanism at this timepoint. Within ~7 days, innate immunity and the IgM response may be dominant antiviral mechanisms. Therefore, DENV2-specific IgM should be measured at the early

timepoint of infection (e.g. Day 3).

Minor critiques

1. Fig.3. Time of infection? 24 hr?
2. Fig.4. D-R. On which day p.i. are these specimens collected for analysis? From A, it is assumed that this is Day 6 p.i. Please clearly indicate.
3. Fig.5 and 6. Again, when are these specimens collected for analysis? Presumably Day 6?
4. The font sizes of the figure labels are way small relative to the figure sizes.

Reply to referees' comments.

Referee #1 (Remarks for Author):

The manuscript by Kishan Gaur et al. described a CXCR3-antagonist, named 7D, that inhibits CXCL4-mediated dengue infection *in vitro*. Also, 7D showed an inhibitory effect on DENV infection in two mouse models. Although the antiviral effects of 7D are obvious, the potential mechanisms are very confusing. There are several major flaws in the mechanism part of this manuscript.

Major concerns:

In vitro study:

1. For the *in vitro* study, 7D only showed inhibitory effect in the presence of CXCL4. How about the impact of 7D on DENV infection without adding of CXCL4? Especially in immune-derived cells, which supposed to express CXCL4 after DENV infection.

Reply: Thank you for your valuable suggestions. We have tested the effect of 7D on CXCL4-expressing cells including mouse primary megakaryocytes and human megakaryoblast cell line MEG-01. 7D inhibited the DENV2 infection in both cell types. These cells secreted elevated CXCL4 in supernatant following DENV2 infection (Fig. EV2). Further, 7D was unable to inhibit viral infection in cells like mouse primary monocytes or human DC-SIGN-U937 cell line that do not secrete CXCL4. Thus, supporting the inhibitory effect of 7D specifically on CXCL4-induced DENV replication, described in the results section, paragraph 3 page 4.

2. 7D may increase IFN expression by the CXCL4: CXCR3: p38:IRF3 pathway. Does 7D also suppress DENV infection in IFN-deficient cells? If that is true, what's the mechanism?

Reply: We have shown that 7D suppressed DENV2 infection in the bone marrow derived macrophages (BMDMs) of AG129 IFN $\alpha/\beta/\gamma$ R^{-/-} mice (Fig. EV6A-L) by stimulating another subset, called type-III or IFN- λ , acting through receptors IFN λ R1 and IL-10R β , and are known for delayed anti-viral response via IRF3 signaling (Wack et al, 2015), described in the result section, paragraph 3 in page 6, as well as in the discussion, paragraph 3 in page 8. Besides, we also described that 7D inhibited viral infection by increasing the synthesis of DENV2-neutralizing IgM/IgG in AG129 IFN $\alpha/\beta/\gamma$ R^{-/-} mice. 7D increased acetylation and phosphorylation of STAT3 in plasma cells, in turn increased the synthesis of antibodies.

3. All the *in vitro* study data are shown as bars, but no spots indicate the number of samples or the number of biological repeats.

Reply: We have mentioned the same in numerical number (n=3) in the Figure legends.

In vivo study:

1. If the antiviral effect of 7D is CXCR3 dependent, the authors should include the experiments using CXCR3^{-/-} mice or using CXCR3 blocking reagent to confirm the results.

Reply: We have performed the experiment using CXCR3^{-/-} mice, described in main Figure 6. Our data describe the inhibition of viral infection by 7D in CXCR3^{-/-} mice by increasing the neutralizing antibody production. 7D supplementation did not alter the IFN α synthesis in these mice. This could be that in absence of CXCR3 axis the stimulatory effects of 7D on IFN axis was impaired. However, it needs further experimental proof. We have described in the results section, paragraph 4 in page 6.

2. AG129 mice lack IFN (I/II) receptors; the elevated IFN levels will have no effect on viral replication in these mice. Thus, the elevated IFNs will not be the mechanism for 7D inhibits DENV in AG129 mice.

Reply: We described above the 7D-mediated stimulation of type-III or IFN- λ , works through receptors IFN λ R1 and IL-10R β , in BMDM of AG129 IFN $\alpha/\beta/\gamma$ R $^{-/-}$ mice. That could a reason behind it.

3. I agree that the elevated Ab titres in 7D- treated mice could be a reason for inhibiting DENV, but why? The authors try to show that 7D increased the Acetylation and phosphorylation of STAT3. But what are the mechanisms behind this? Is it CXCR3 dependent? Also, in CXCR3 $^{-/-}$ mice, does 7D increase DENV-induced antibody production?

Reply: We described in our above reply that as a Sirt-1 inhibitor, 7D increases the acetylation and phosphorylation of STAT3 in plasma cells, in turn increases the synthesis of both IgM and IgG antibodies, and enhances viral neutralization. We examined the same in CXCR3 $^{-/-}$ mice and observed the increased in percentage of plasma cells and elevated antibodies in circulation of these mice after 7D treatment. Importantly, we observed elevated acetylation and phosphorylation of STAT3 in the leukocytes isolated from the spleen of these mice, included as main Fig. 6, and described in the results section, paragraph 4 in page 6.

4. The authors cited a reference to show 7D may increase the Acetylation of STAT3 by inhibiting Sirt1 but did not show any data on this during DENV infection. Does 7D inhibit DENV infection in Sirt1 or STAT3-deficient mice?

Reply: We could get the SIRT1 $^{-/-}$ mouse embryonic fibroblasts (MEF's) as the kind gift from Dr Ullas S Kolthur, Centre for DNA Fingerprinting & Diagnostics, Hyderabad, India; and Dr. Michale Mcburney, Ottawa Hospital Research institute, Canada, Kolthur-Seetharam et al, 2006). Further, we performed the above experiment and observed the inhibitory effects of 7D on DENV2 infection on both Sirt1 $^{-/-}$ and WT MEFs (Fig. EV3B-M). Although 7D could decrease the protein acetylation in WT but not in Sirt1 $^{-/-}$ MEFs. Thus, ruling out a direct involvement of 7D:Sirt1 axis in DENV2 replication in vitro, described in the results section, paragraph 4 in page 4. However, as described above, 7D supplementation increases the acetylation of STAT3 and stimulates plasma cell proliferation, in turn antibody secretion in mice in vivo.

Generally, the two mechanisms for 7D inhibit DENV in vivo are very unclear. There is a lot of work to do before the manuscript can be accepted for publication.

Reply: We have performed new experiments and accordingly modified the results and discussion section to clarify the mechanisms.

Referee #2 (Comments on Novelty/Model System for Author): See comments to author.

Referee #2 (Remarks for Author):

This is a comprehensive study that identifies a novel antagonist of CXCR3, through in silico screening, followed by testing in various in vitro and in vivo systems to demonstrate the small molecule is effective in modulating signalling to dengue infection and that it can enhance interferon production in vivo, while reducing infection. The authors also characterize the effects of the drug on cellular immune responses and antibodies following infection. The use of CXCR3-KO cells to demonstrate the mechanism of drug is through this receptor is an elegant supporting study. The use of both IFN deficient and WT animals is also an advantage of the study since the results were consistent. The read-outs in vivo are quite broad, and more will need to be done (in the future, not necessarily for this first manuscript) to define the mechanisms through which the drug works in vivo to alter T cell and antibody responses. Although some signalling is presented, I am not sure this is the whole mechanism, and the potential for off-target effects were not thoroughly investigated, so perhaps some discussion of those points would improve the manuscript.

Specific suggestions for improvements:

Perhaps a draw-back of the study is that it is not necessarily shown that reduced TNF expression is beneficial in dengue. Although TNF has been associated with vascular leak in some models, the use of T cell derived TNF has not been explicitly investigated. It should also be noted that TNF has beneficial roles in adaptive immune responses aside from inducing vascular permeability.

Reply: Thank you for your valuable suggestions. We have removed the part of TNF positive T cells data from the Figure (Figure 5 in earlier version) and text. We agree with you that it needs a further detailed instigation, how these cells are associated with vascular leakage in one hand, and adaptive immune responses on the other hand. In fact, removing these parts do not dilute the main objectives of this study.

The authors should check that the word "significantly" is used to mean statistical significance only rather than to highlight compelling conclusions.

Reply: We have corrected the same in whole text.

I am not sure if the title is clear what 7D is- Maybe the words "small molecule" could be added to the title?

Reply: Absolutely right, we have included in the title, as well as in text.

This statement needs a citation: "The increased CD4CD8 double positive T cells is considered to be associated with the severity of dengue." It is a bit unexpected that the authors chose to focus on double positive cells. Panels 5A-B should be described in the text as well. It's not specifically stated that those populations did not differ.

Reply: Please find our above reply to your comments.

For the figures, all of the labels - e.g. A, B, C are quite small. Figure 3 would also benefit from aligning the panels more accurately.

The figure 4 legend should indicate which panels were performed in WT mice, since this figure apparently contains both data from AG129 and WT mice.

Reply: We have aligned the Fig. 3. Data in Fig. 4 is completely from AG129 mice experiment. We have highlighted it in the Figure legend.

Referee #3 (Comments on Novelty/Model System for Author): AG129 and WT mouse models are fine
Referee #3 (Remarks for Author):

In this manuscript titled "7D inhibits dengue infection by increasing synthesis of IFN α/β and neutralizing antibodies via CXCL4:CXCR3:p38:IRF3 and Sirt1:STAT3 axes respectively", Kishan Kumar Gaur et al. identify a CXCR3 inhibitor by in silico modelling, i.e., 7D. They show that 7D inhibits CXCL4-enhanced DENV2 replication in U937-DC-SIGN cell and primary mouse monocytes. Using IFN $\alpha/\beta/\gamma$ R-/-AG129 and WT mice, the authors show that 7D is safe and potent to suppress DENV2 replication, it renders animals resistant to lethal DENV2 infection in AG129 mice, reduces vascular leakage, thrombocytopenia, and leukopenia. Mechanistically, 7D promotes type I IFN expression by suppressing the CXCL4:CXCR3:p38:IRF3 pathway, enhances STAT3-acetylation in plasma cells thus neutralizing antibodies. Another potent CXCR3 inhibitor, AMG 487, promotes type I IFN expression, protects AG119 mice from DENV2 infection, but has no effect on B cells and neutralizing antibodies.

Overall, the study identifies a new CXCR3 inhibitor that might be therapeutic against DENV2 infection. However, the mechanism of action of 7D requires more rigorous and direct evidence.

Major critiques

1. The fact that AMG487 still inhibits DENV2 replication in AG129 mice without increasing neutralizing antibodies suggests that the primary antiviral mechanism of AMG487 is independent of the type I IFN response. Different from AMG487, 7D promotes both type I IFN and neutralizing antibody production, the latter may inhibit DENV replication in AG129 mice in the absence of IFN-I signalling. The authors need to strengthen that 7D and AMG487 indeed inhibits DENV2 via type I IFNs using AG129 monocytes *ex vivo*.

Reply: Thank you for your valuable comments. We have performed new experiments and described that AMG487 supplementation stimulated another IFN subset, called type-III or IFN- λ , works through receptors IFN λ R1 and IL-10R β , in the bone marrow derived macrophages (BMDMs) of AG129 IFN $\alpha/\beta/\gamma$ R $^{-/-}$ mice (Fig. EV6A-L). Besides, our molecule 7D also showed the similar mechanism of stimulating IFN- λ . The type III IFNs, which are known for delayed type anti-viral responses, and are regulated by IRF3 signaling (Wack *et al*, 2015), described in the result section, paragraph 3 in page 6, as well as in the discussion, paragraph 3 in page 8. This could be a possible reason for the reduced viral load in IFN $\alpha/\beta/\gamma$ R $^{-/-}$ BMDM upon AMG487 treatment.

2. Small molecule inhibitors may have many off-target effects *in vivo*. The authors should provide direct evidence that CXCR3 signalling facilitates DENV2 and that 7D inhibits DENV2 infection via CXCR3 *in vivo*, since the Cxcr3 $^{-/-}$ animals are available (Fig.2). Are Cxcr3 $^{-/-}$ animals resistant to DENV2 infection and is the effect of 7D lost in Cxcr3 $^{-/-}$ mice? Do Cxcr3 $^{-/-}$ mice have more neutralizing antibodies?

Reply: We examined the same in CXCR3 $^{-/-}$ mice and observed the increased percentage of plasma cells and elevated antibodies (IgM and IgG) in circulation of these mice after 7D treatment. Importantly, we observed elevated acetylation and phosphorylation of STAT3 in B lymphocytes isolated from the spleen of these mice, described in the results section, paragraph 4 in page 6, and main Figure 6.

3. Fig.6. It is not clear about the time of sampling, presumably Day 6 p.i. based on Fig.4A. This reviewer is not sure DENV2-specific IgG is significantly induced and is the primary antiviral mechanism at this timepoint. Within ~7 days, innate immunity and the IgM response may be dominant antiviral mechanisms. Therefore, DENV2-specific IgM should be measured at the early timepoint of infection (e.g. Day 3).

Reply: We have corrected the same, 6DPI, in the Figure legends. We agree with referee's comments. We measured the IgM levels in mice plasma from 3DPI, and observed the elevated IgM in plasma of the mice treated with 7D, included as Fig. 5A. We have included in the result section.

Minor critiques

1. Fig.3. Time of infection? 24 hr?

Reply: In Fig. 3, *in vitro* experiments were terminated at 24 hrs. We have mentioned in the Figure legend.

2. Fig.4. D-R. On which day p.i. are these specimens collected for analysis? From A, it is assumed that this is Day 6 p.i. Please clearly indicate.

Reply: We feel sorry for the confusion. All mice experiments were terminated at 6DPI and we have highlighted in the same in the Figure legends.

3. Fig.5 and 6. Again, when are these specimens collected for analysis? Presumably Day 6?

Reply: Please find our above reply to your comments.

4. The font sizes of the figure labels are way small relative to the figure sizes.

Reply: We have changed.

10th Jun 2024

Dear Dr. Guchhait,

Thank you again for submitting your revised work to EMBO Molecular Medicine. We have now heard back from the original three reviewers who evaluated your study. As you will see below, the reviewers are supportive on the potential therapeutic benefits of 7D but still have remaining concerns on its mechanism. We would therefore ask you to address their concerns in a revision. In particular it will be necessary to tone down the conclusions and highlight more clearly the benefit against DENV infection while discussing the limitations on the mechanistic basis. In addition, we would encourage you to address some of Reviewer 1's points in order to strengthen some aspects of the mechanism and further improve the manuscript.

We remind you that we have the following formatting requirements:

4) A .docx formatted letter INCLUDING the reviewers' reports and your detailed point-by-point responses to their comments. As part of the EMBO Press transparent editorial process, the point-by-point response is part of the Review Process File (RPF), which will be published alongside your paper.

5) A complete author checklist, which you can download from our author guidelines (<https://www.embopress.org/page/journal/17574684/authorguide#submissionofrevisions>). Please insert information in the checklist that is also reflected in the manuscript. The completed author checklist will also be part of the RPF.

6) Please note that all corresponding authors are required to supply an ORCID ID for their name upon submission of a revised manuscript.

7) It is mandatory to include a 'Data Availability' section after the Materials and Methods. Before submitting your revision, primary datasets produced in this study need to be deposited in an appropriate public database, and the accession numbers and database listed under 'Data Availability'. Please remember to provide a reviewer password if the datasets are not yet public (see <https://www.embopress.org/page/journal/17574684/authorguide#dataavailability>).

This study includes no data deposited in external repositories.

8) For data quantification: please specify the name of the statistical test used to generate error bars and P values, the number (n) of independent experiments (specify technical or biological replicates) underlying each data point and the test used to calculate p-values in each figure legend. The figure legends should contain a basic description of n, P and the test applied. Graphs must include a description of the bars and the error bars (s.d., s.e.m.). Please provide exact p values.

13) Author contributions: CRediT has replaced the traditional author contributions section because it offers a systematic machine readable author contributions format that allows for more effective research assessment. Please remove the Authors Contributions from the manuscript and use the free text boxes beneath each contributing author's name in our system to add specific details on the author's contribution. More information is available in our guide to authors.

Please also suggest a visual abstract to illustrate your article as a jpeg file 550 px wide x 300-600 px high.

Share synopsis text and image, as well as eTOC:

Please note that these would be the final versions and changes during proofing are usually not allowed

16) As part of the EMBO Publications transparent editorial process initiative (see our policy here: https://www.embopress.org/transparent-process#Review_Process), EMBO Molecular Medicine will publish online a Peer Review File (PRF) to accompany accepted manuscripts.

In the event of acceptance, this file will be published in conjunction with your paper and will include the anonymous referee reports, your point-by-point response and all pertinent correspondence relating to the manuscript. Let us know whether you agree with the publication of the PRF and as here, if you want to remove or not any figures from it prior to publication.

Yours sincerely,

Poonam Bheda

Poonam Bheda, PhD
Scientific Editor
EMBO Molecular Medicine

***** Reviewer's comments *****

Referee #1 (Comments on Novelty/Model System for Author):

The molecular mechanism part is pretty weak in this study.

Referee #1 (Remarks for Author):

I am still not satisfied with the authors' explanation of the mechanism by which 7D inhibits the viral response in vivo.

1, The authors claim that type III IFN may play a role, yet how does 7D activate type III interferon expression? How do you prove that it is through the type III IFN signaling pathway? Are there any experiments blocking this pathway in AG129-derived cells or mice to verify this?

BTW , the mechanism by which 7D promotes IFN expression has been very poorly studied.

2, The evidence that 7D regulates antibody production through STAT3 is still insufficient. Can an altered state of STAT3 acetylation phosphorylation in leukocytes (in B or T cells?) determine the antibody levels?

Would 7D still cause an increase in antibody titers in the presence of other STAT3 inhibitors?

Can 7D still be antiviral in the presence of STAT3 inhibitors?

Referee #2 (Comments on Novelty/Model System for Author):

Numerous knock-out mice were used to identify multiple steps of the drug's mechanism of action.

Referee #2 (Remarks for Author):

The authors addressed all of my comments. I think that the manuscript has been improved. I would highlight they have found multiple possible mechanisms of action but this does not detract from the interest of the manuscript since real drugs do work via multiple pathways often. I think this is sufficient for the first report and the authors have improved the wording for the conclusions. There could be follow up studies to identify and confirm other aspects of the antiviral activity, such as what mechanisms might occur in various other knock out mice, in my view.

Referee #3 (Comments on Novelty/Model System for Author):

The mouse model is fine.

Referee #3 (Remarks for Author):

The authors have largely addressed my previous critiques. I now have two minor comments. First, the new data have shown that CXCR3 signaling also inhibits type III IFNs, the title and Figure 7 may need updating. Second, in Figure 6, the authors performed DENV infection in Cxcr3^{-/-} mice. This should have been compared to WT mice in parallel. Nonetheless, I understand that neither Cxcr3 knockout mice nor WT adult mice succumb to DENV infection. The authors should at least make it clear to readers that Cxcr3^{-/-} adult mice don't succumb to DENV in either the result or discussion section.

Reply to Referee

Referee #1 (Remarks for Author):

I am still not satisfied with the authors' explanation of the mechanism by which 7D inhibits the viral response in vivo.

1, The authors claim that type III IFN may play a role, yet how does 7D activate type III interferon expression? How do you prove that it is through the type III IFN signaling pathway? Are there any experiments blocking this pathway in AG129-derived cells or mice to verify this?

BTW, the mechanism by which 7D promotes IFN expression has been very poorly studied.

Reply: Thank you for valuable suggestion. We have performed the experiment in vitro using bone marrow derived macrophages (BMDMs) of AG129 mice, described in a new Figure EV4. These macrophages secreted detectable amount of IFN- λ . 7D treatment increased the IFN- λ level and decreased the DENV2 load. A blocking antibody to IFN- λ 2/3 inhibited the 7D-mediated IFN- λ functions, in turn decreased the IFN-stimulated genes (ISGs) and increased DENV2 load. 7D uses CXCL4: CXCR3: p38: IRF3 pathway to regulate IFN- α/β as well as IFN- λ synthesis, explained in the results and discussion sections.

Although 7D increased IFN- $\alpha/\beta/\lambda$ levels, it did not alter IFN γ level significantly. We described in last paragraph of discussion section as caveat that needs further detailed studies.

2, The evidence that 7D regulates antibody production through STAT3 is still insufficient. Can an altered state of STAT3 acetylation phosphorylation in leukocytes (in B or T cells?) determine the antibody levels?

Would 7D still cause an increase in antibody titers in the presence of other STAT3 inhibitors?

Can 7D still be antiviral in the presence of STAT3 inhibitors?

Reply: Thank you for this suggestion. We used STAT3 inhibitor Stattic in this study. AG129 mice were infected with DENV2 and were supplemented with 7D alone, 7D+Stattic or Stattic alone. Data are included as a new Figure EV5.

Yes, STAT3 acetylation and phosphorylation determine the plasmablast proliferation and antibody generation. 7D treatment increased antibody levels. Stattic treatment suppressed the 7D-mediated elevation in antibody level, by suppressing phosphorylation STAT3^{Y705} but not acetylation STAT3^{K685}. Data also indicated that STAT3^{K685} regulates STAT3^{Y705} activation following 7D treatment.

Although, we could explain that 7D: Sirt1: STAT3^{K685}: STAT3^{Y705} axis is an important regulator of plasmablasts and antibody generation. The above observation has limitation in explaining the decrease in both viral load as well as neutralizing antibodies against DENV2 in Stattic-treated mice. It may be due to a global anti-viral effect of this STAT3-inhibitor, which needs further investigations.

We described in the results and discussion sections.

Referee #2 (Remarks for Author):

The authors addressed all of my comments. I think that the manuscript has been improved. I would highlight they have found multiple possible mechanisms of action

but this does not detract from the interest of the manuscript since real drugs do work via multiple pathways often. I think this is sufficient for the first report and the authors have improved the wording for the conclusions. There could be follow up studies to identify and confirm other aspects of the antiviral activity, such as what mechanisms might occur in various other knock out mice, in my view.

Reply: Thank you for valuable suggestion. We have highlighted the anti-viral effects of this small molecule in whole text. Also we explained the possible mechanisms.

Referee #3 (Remarks for Author):

The authors have largely addressed my previous critiques. I now have two minor comments. First, the new data have shown that CXCR3 signaling also inhibits type III IFNs, the title and Figure 7 may need updating. Second, in Figure 6, the authors performed DENV infection in *Cxcr3*^{-/-} mice. This should have been compared to WT mice in parallel. Nonetheless, I understand that neither *Cxcr3* knockout mice nor WT adult mice succumb to DENV infection. The authors should at least make it clear to readers that *Cxcr3*^{-/-} adult mice don't succumb to DENV in either the result or discussion section.

Reply: Thank you for valuable suggestion. We have made the changes in the title and Fig. 7.

We have included WT and CXCR3^{-/-} mice data in the Fig.6.

We have described in the discussion that 7D treatment increased the neutralizing antibodies via Sirt1:STAT3 signaling that provided protection to the CXCR3^{-/-} mice against dengue infection, even they do not have a functional 7D: CXCR3:IFN axis.

15th Aug 2024

Dear Dr. Guchhait,

Thank you for the submission of your revised manuscript to EMBO Molecular Medicine. Your manuscript has now been re-reviewed by one of the original reviewers. Based on their advice (included below), I am pleased to inform you that we will be able to accept your manuscript pending the following final amendments:

1) In a routine check for text similarity, we note that the majority of sentences in the Introduction are almost exactly copied from previous reviews from your lab, including Singh, *Front. Cell. Infect. Microbiol* 2020 and Chauhan, *Bioscience Reports* 2024. Please rewrite the Introduction such that the text is significantly different from both of these previous publications.

2) In the main manuscript file, please reduce keywords to max. 5.

3) We require genome sequencing datasets to be uploaded to an appropriate repository (INSDC approved) and made publicly available prior to publication. Please upload the viral genome sequencing data to a repository and include the information to access the data in the Data availability section formatted according to the example below:

"The datasets and computer code produced in this study are available in the following databases:

- Chip-Seq data: Gene Expression Omnibus GSE46748 (<https://www.ncbi.nlm.nih.gov/geo/query/acc.cgi?acc=GSE46748>)

- Modeling computer scripts: GitHub (<https://github.com/SysBioChalmers/GECKO/releases/tag/v1.0>)

- [data type]: [full name of the resource] [accession number/identifier] ([doi or URL or identifiers.org/DATABASE:ACCESSION])"

4) Please rename "Competing Interests" to "Disclosure and competing interests statement". We updated our journal's competing interests policy in January 2022 and request authors to consider both actual and perceived competing interests. Please review the policy <https://www.embopress.org/competing-interests> and update your competing interests if necessary.

5) Author contributions: Please remove it from the manuscript and specify author contributions in our submission system. CRediT has replaced the traditional author contributions section because it offers a systematic machine-readable author contributions format that allows for more effective research assessment. You are encouraged to use the free text boxes beneath each contributing author's name to add specific details on the author's contribution. More information is available in our guide to authors:

<https://www.embopress.org/page/journal/17574684/authorguide#authorshipguidelines>

6) In the Methods, please take care of the following:

- Studies with human research participants: The use of human samples requires information on the authority granting ethics approval (e.g. IRB) and informed consent. If the need for approval is waived, please cite the reason (e.g. non-human subject research because the samples used were de-identified/coded with no identifying information) and legislation in the relevant methods section.

- If the study did in fact have human research participants with approval, please also state that the experiments conformed to the principles set out in the WMA Declaration of Helsinki and the Department of Health and Human Services Belmont Report. Please note that this is a separate statement from the specific ethics committee approval and informed consent.

- Animals: Please ensure that housing and husbandry conditions and gender of animals involved in experiments is reported as these are not currently reported.

- Cell lines: Please include all information requested in the author checklist for cell lines used in the manuscript (accession number in repository or supplier name, catalog number, clone number, and/or RRID) Currently catalog/clone numbers are missing. Please also be sure to include a sentence in the Methods as to whether or not the cell lines were recently authenticated and tested for mycoplasma contamination.

- Please ensure that a statement on whether or not blinding was done is included in the Methods even if no blinding was done.

- Antibodies: please ensure that company name, catalog number, and dilutions/amounts of each antibody are reported. Currently most of this information is missing throughout the Methods section.

7) All materials and methods need to be described in the main text using our 'Structured Methods' format, which is required for all research articles. According to this format, the Methods section includes a Reagents and Tools Table (listing key reagents, experimental models, software and relevant equipment and including their sources and relevant identifiers) followed by a Methods and Protocols section describing the methods using a step-by-step protocol format. The aim is to facilitate adoption of the methodologies across labs. More information on how to adhere to this format as well as a downloadable template (.docx) for the Reagents and Tools Table can be found in our author guidelines:

<https://www.embopress.org/page/journal/17574684/authorguide#structuredmethods>

8) Please place individual sections of the manuscript in the following order: Title page - Abstract & Keywords - Introduction - Results - Discussion - Methods - Data Availability - Acknowledgements - Disclosure and Competing Interests Statement - The Paper Explained - References - Figure Legends - Expanded View Figure Legends.

9) For the figures and figure legends, please take care of the following:

- Please make sure to update the callout for Table S2 to 'Appendix Table S2'.

- Please note that a separate 'Data Information' section is required in the legends of figures 3a-c, e-g; EV 2a-h; EV 6b-l.

- Please note that the legend for figures 1g-h have been interchanged. This needs to be rectified.
- Please note that the legend for figure 1g is incorrectly labelled as 1h in the "data information." This needs to be rectified.
- Please note that the legend for figure 4q-r is incorrectly labelled as 4p-r. This needs to be rectified.
- Please define the annotated p values *** in the legend of figure 4b; as appropriate.
- Please indicate the statistical test used for data analysis in the legends of figures 4d; EV 1f; EV 3a, j-k; EV 4b-c; EV 5a-b.
- Please note that information related to n is missing in the legends of figures 2d-e; 4e-m, o, q-r; 5c-e, g-j, l-m, o; 6d-i; EV 1d; EV 2a-h; EV 3a, j-m; EV 4a-c, h-i; EV 5a-b, f-i; EV 6d;
- Although 'n' is provided, please describe the nature of entity for 'n' in the legends of figures 1i; 2h-k; 3a-c, e-g; EV 1k-l; EV 6b-c.
- Please note that the error bars are not defined in the legends of figures 1f; 4q-r; 5c, h-j; 6g; EV 1d, EV 5h.
- Please note that we require exact p-values to be reported in either the figure or figure legend. Currently exact p-values are not provided.

10) Appendix file: Please add page numbers to the Table of Contents.

11) Funding: Please ensure that all funding sources are entered into the manuscript submission system. Currently Department of Science and Technology, Govt. of India for DST-INSPIRE Fellowship (2014/113) is missing.

12) Synopsis:

- Synopsis image: Please provide a graphic that summarises the main findings of the manuscript on a glance and upload it as a high-resolution jpeg file 550 pixels wide x (250-400) pixels high.

- Synopsis text: Please reformat the synopsis text to include a short standfirst (maximum of 300 characters, including space) followed up by to 5 bullet points that summarise the key new findings (maximum of 30 words / bullet point) Please use the passive voice.

13) The Paper Explained: Please include "The Paper Explained" in the main manuscript text and not as a separate file.

14) As part of the EMBO Publications transparent editorial process initiative (see our policy here:

https://www.embopress.org/transparent-process#Review_Process), EMBO Molecular Medicine will publish online a Peer Review File (PRF) to accompany accepted manuscripts. This file will be published in conjunction with your paper and will include the anonymous referee reports, your point-by-point response and all pertinent correspondence relating to the manuscript. Let us know whether you agree with the publication of the PRF and as here, if you want to remove or not any figures from it prior to publication. Please note that the Authors checklist will be published at the end of the PRF.

15) Please provide a point-by-point letter INCLUDING my comments as well as the reviewer's reports and your detailed responses (as Word file).

I look forward to reading a new revised version of your manuscript as soon as possible.

Yours sincerely,

Poonam Bheda

Poonam Bheda, PhD
Scientific Editor
EMBO Molecular Medicine

***** Reviewer's comments *****

Referee #1 (Remarks for Author):

I agree with Reviewer 2's opinion that although there may be many possibilities in the mechanism parts, the effect of the drug is evident. I agree to publish this work and suggest that the author further refine the detailed mechanisms in future work.

The authors addressed the minor editorial issues.

27th Aug 2024

Dear Dr. Guchhait,

Thank you for the submission of your revised manuscript to EMBO Molecular Medicine. There continue to be formatting requests that we would ask you to address in a revision as follows:

- As previously pointed out, there continue to be sentences very similar to Chauhan, Bioscience Reports 2024. Please edit the first sentence of the abstract further to distinguish from the abstract of your previous publication. Please also edit the sentence "...sub-neutralizing cross-reactive antibodies opsonize...". Finally we would suggest that you edit the sentence "...viral disease casue by the dengue virus..." in the introduction as it also contains significant similarity to a thesis available online.
- As previously requested, please include accession numbers or catalog/clone numbers for each cell line used in the Methods.
- Please also be sure to include a sentence in the Methods as to whether or not the cell lines were recently authenticated.
- As previously requested, please ensure that specific catalog numbers for each antibody is reported throughout the Methods section.
- Although 'n' is provided, please describe the nature of entity for 'n' in the legends of figures 1i; 2h-k; 3e-g; EV 1k-l
- The error bars are not defined in the legend of Figure 5j

I look forward to reading a new revised version of your manuscript as soon as possible. Please let me know if you have any questions.

Yours sincerely,

Poonam Bheda

Poonam Bheda, PhD
Scientific Editor
EMBO Molecular Medicine

The authors addressed the remaining editorial issues.

Reply to the Editor's comments

- As previously pointed out, there continue to be sentences very similar to Chauhan, Bioscience Reports 2024. Please edit the first sentence of the abstract further to distinguish from the abstract of your previous publication. Please also edit the sentence "...sub-neutralizing cross-reactive antibodies opsonize...". Finally we would suggest that you edit the sentence "...viral disease casue by the dengue virus..." in the introduction as it also contains significant similarity to a thesis available online.

Reply. Thank you Editor for your time and comments. We are sorry for these mistakes. We have included the corrections. After corrections, we have performed the similarity check.

- As previously requested, please include accession numbers or catalog/clone numbers for each cell line used in the Methods.

Reply. We have included the catalog/accession numbers for cell lines in the Methods section.

- Please also be sure to include a sentence in the Methods as to whether or not the cell lines were recently authenticated.

Reply. We have included the sentence in paragraph 1 page 11.

- As previously requested, please ensure that specific catalog numbers for each antibody is reported throughout the Methods section.

Reply. We have included the catalog numbers for antibodies in the Methods section.

- Although 'n' is provided, please describe the nature of entity for 'n' in the legends of figures 1i; 2h-k; 3e-g; EV 1k-l

- The error bars are not defined in the legend of Figure 5j

Reply. We have included the above corrections.

29th Aug 2024

Dear Dr. Guchhait,

We are pleased to inform you that your manuscript is accepted for publication and is now being sent to our publisher to be included in the next available issue of EMBO Molecular Medicine.

Yours sincerely,

Poonam Bheda, PhD
Scientific Editor
EMBO Molecular Medicine
